# Structural Analysis and Reactivity Insights of (*E*)-Bromo-4-((4-((1-(4-chlorophenyl)ethylidene)amino)-5-phenyl-4H-1,2,4-triazol-3-yl)thio)-5-((2-isopropylcyclohexyl)oxy)Furan-2(5H)-one: A Combined Approach Using Single-Crystal X-ray Diffraction, Hirshfeld Surface Analysis, and Conceptual Density Functional Theory

**Ahmed H. Bakheit** *, **Mohamed W. Attwa** *, **Adnan A. Kadi** and **Hamad M. Alkahtani**

Department of Pharmaceutical Chemistry, College of Pharmacy, King Saud University,
P.O. Box 2457, Riyadh 11451, Saudi Arabia; akadi@ksu.edu.sa (A.A.K.); ahamad@ksu.edu.sa (H.M.A.)
* Correspondence: abakheit@ksu.edu.sa (A.H.B.); mzeidan@ksu.edu.sa (M.W.A.); Tel.: +966-1146-97673 (A.H.B.); +966-1146-70237 (M.W.A.); Fax: +966-1146-76220 (M.W.A.)

**Abstract:** This study presents a comprehensive exploration of the structure–reactivity relationship of (*E*)-3-bromo-4-((4-((1-(4-chlorophenyl)ethylidene)amino)-5-phenyl-4H-1,2,4-triazol-3-yl)thio)-5-((2-isopropylcyclohexyl)oxy)furan-2(5H)-one. The study embarked on an in-depth investigation into the solid-state crystal structure of this organic compound, employing computational Density Functional Theory (DFT) and related methodologies, which have not extensively been used in the examination of such compounds. A single-crystal X-ray diffraction (SCXRD) analysis was initially performed, supplemented by a Hirshfeld surfaces analysis. This latter approach was instrumental in visualizing and quantifying intermolecular interactions within the crystal structures, offering a detailed representation of the molecule's shape and properties within its crystalline environment. The concept of energy framework calculations was utilized to understand the varied types of energies contributing to the supramolecular architecture of the molecules within the crystal. The Conceptual DFT (CDFT) was applied to predict global reactivity descriptors and local nucleophilic/electrophilic Parr functions, providing a deeper understanding of the compound's chemical reactivity properties. The aromatic character and π–π stacking ability were also evaluated with the help of LOLIPOP and ring aromaticity measures. This comprehensive approach not only provides a detailed description of the structure and properties of the investigated compound but also offers valuable insights into the design and development of new materials involving 1,2,4-triazole systems.

**Keywords:** 1,2,4-triazole derivatives; density functional theory (DFT); Hirshfeld surface analysis; global reactivity descriptors; local Parr functions; aromatic character; π–π stacking ability; intermolecular interactions; frontier molecular orbitals (FMOs)

## 1. Introduction

Heterocyclic organic compounds, such as 1,2,4-triazole derivatives, are recognized for their structural diversity and selective chemical reactivity. The molecules within this class have been the subject of considerable attention due to their broad spectrum of biological properties, including antibacterial, antifungal, anticonvulsant, anti-inflammatory, anticancer, and anti-proliferative activities. The 1,2,4-triazole nucleus has been integrated into a wide range of therapeutically beneficial molecules, improving their drug efficacy. Additionally, Schiff bases of 1,2,4-triazoles have demonstrated substantial biological activities [1].

Schiff bases incorporating 1,2,4-triazole moieties exhibit a broad spectrum of applications, predominantly due to their pronounced biological activities. Specifically, Schiff bases

synthesized from 3-substituted-4-amino-5-mercapto-1,2,4-triazoles have demonstrated potential in analgesic, antimicrobial, anti-inflammatory, and antidepressant therapeutic areas [2,3].

In this study, we aim to delve into the structure–reactivity relationship of (*E*)-3-bromo-4-((4-((1-(4-chlorophenyl)ethylidene)amino)-5-phenyl-4H-1,2,4-triazol-3-yl)thio)-5-((2-isopropylcyclohexyl)oxy) furan-2(5H)-one, a member of the 1,2,4-triazole Schiff base cyclic class. In addition, the successful synthesis of the tested compound was accomplished, as depicted in Scheme 1 [1]. To our knowledge, the solid-state crystal structure of such a family of organic compounds has yet to be comprehensively examined using the computational Density Functional Theory (DFT) and related methodologies.

**Scheme 1.** Depiction of molecular structures and pathway for regioselective synthesis.

Our investigation began with a single-crystal X-ray diffraction (SCXRD) analysis, underpinned by Hirshfeld surface analysis approaches. This method [4], a potent tool for visualizing and quantifying intermolecular interactions in crystal structures [5,6], was employed. This method offers an intricate representation of the shape and properties of a molecule within its crystalline environment. Additionally, we utilized the concept of "fingerprinting" [7] intermolecular interactions [8,9] and energy framework calculations to understand the various types of energies contributing to the supramolecular architecture of molecules within the crystal [10,11].

Utilizing Conceptual DFT (CDFT), we predicted global reactivity descriptors and local Parr functions for nucleophilic/electrophilic tendencies, providing a deeper insight into the chemical reactivity properties. The aromatic character and π–π stacking ability were also evaluated using LOLIPOP and ring aromaticity measures. Through this comprehensive approach, we hope to offer not just a detailed description of the structure and properties of the investigated compound but also valuable insights for synthetic organic chemists to design and develop new materials involving 1,2,4-triazole systems [12,13].

## 2. Materials and Methods

### 2.1. Single-Crystal X-ray Diffraction

The methodology of single-crystal X-ray diffraction (SCXRD) to confirm the stereochemistry of tested compounds involved several steps. The chemical preparations of these compounds were previously reported by Li et al. [1] (CCDC 829447).

In this analysis, the crystalline structures of the compounds were fully described based on the SCXRD results. The intensity data were measured at room temperature (293 K) using a SMART APEX BRUKER AXS diffractometer (Billerica, MA, USA). This instrument is equipped with graphite monochromated MoKα radiation (λ = 0.71073 Å) and operated in the ω/2θ scan mode [1].

The structure solutions for the compound were obtained through direct methods. These were accomplished using the Olex2-1.5-alpha environment [14], Mercury 4.0 [15], and the computer program SHELXT [16]. Details of the structure determination, including crystal data, are summarized in Table 1. The geometrical calculations and weak interaction identifications were generated using the ORTEP3 program [17]. Finally, selected bond distances and angles were collected and are detailed in Tables 2 and 3 for each tested compound.

**Table 1.** Weak bond geometry (Å).

| Number | Atom1 | Atom2 | Length | Length-VdW | Rv | Symm. op. 1 [a] | Symm. op. 2 [b] |
|--------|-------|-------|--------|------------|-----|----------------|-----------------|
| 1 | O1 | H1 | 2.454 | −0.266 | 9.78 | x, y, z | x, y, −1 + z |
| 2 | H16B | Cl1 | 2.936 | −0.014 | 0.47 | x, y, z | −x, −1 + y, −z |
| 3 | C10 | H5 | 2.881 | −0.019 | 0.66 | x, y, z | 1/2 − x, −1/2 + y, −z |
| 4 | C12 | N1 | 3.184 | −0.066 | 2.03 | x, y, z | 1/2 − x, −1/2 + y, −z |
| 5 | H12 | N1 | 2.43 | −0.32 | 11.64 | x, y, z | 1/2 − x, −1/2 + y, −z |

[a&b] "Symm. op. 1" and "Symm. op. 2" columns describe the symmetry operations that relate the two interacting atoms as part of the crystal's symmetry. These operations are used to generate the entire crystal structure from a single asymmetric unit.

**Table 2.** Comparative analysis of selected bond lengths (Å) for the investigated compound as determined by SCXRD and DFT methods.

| Atom–Atom | Length/Å | | | Atom–Atom | Length/Å | | |
|-----------|----------|-----|-------------|-----------|----------|-----|-------------|
| | SCXRD | DFT | \|SCXRD-DFT\| | | SCXRD | DFT | \|SCXRD-DFT\| |
| Br1–C10 | 1.869(5) | 1.869 | 0.0007 | C13–C18 | 1.510(6) | 1.51 | 0.0003 |
| C1–C2 | 1.378(8) | 1.378 | 0.001 | C13–O3 | 1.466(6) | 1.466 | 0.0008 |
| C1–C6 | 1.397(7) | 1.397 | $1 \times 10^{-4}$ | C14–C15 | 1.509(7) | 1.509 | 0 |
| C2–C3 | 1.363(8) | 1.363 | 0 | C15–C16 | 1.530(8) | 1.53 | 0.0011 |
| C3–C4 | 1.378(8) | 1.378 | 0.0009 | C15–C22 | 1.515(9) | 1.515 | 0.0005 |
| C4–C5 | 1.362(7) | 1.362 | 0.0009 | C16–C17 | 1.518(9) | 1.518 | 0.0012 |
| C5–C6 | 1.389(7) | 1.389 | 0.0004 | C17–C18 | 1.526(7) | 1.526 | $1 \times 10^{-4}$ |
| C6–C7 | 1.486(7) | 1.486 | 0.0002 | C18–C19 | 1.548(7) | 1.548 | $1 \times 10^{-4}$ |
| C7–N1 | 1.315(5) | 1.315 | 0.0002 | C19–C20 | 1.517(8) | 1.517 | 0.0002 |
| C7–N3 | 1.336(6) | 1.336 | 0.0003 | C19–C21 | 1.527(7) | 1.527 | 0.0003 |
| C8–N2 | 1.310(6) | 1.31 | $1 \times 10^{-4}$ | C23–C24 | 1.464(7) | 1.464 | $1 \times 10^{-4}$ |
| C8–N3 | 1.382(6) | 1.382 | 0.0003 | C23–N4 | 1.261(6) | 1.261 | 0.0006 |
| C8–S1 | 1.743(5) | 1.743 | 0.0003 | C24–C25 | 1.339(8) | 1.339 | 0.0004 |
| C9–C10 | 1.329(6) | 1.329 | 0.0004 | C24–C29 | 1.369(8) | 1.369 | $1 \times 10^{-4}$ |
| C9–C12 | 1.516(6) | 1.516 | 0.0006 | C25–C26 | 1.397(8) | 1.397 | 0.0005 |
| C9–S1 | 1.735(5) | 1.735 | 0.0004 | C26–C27 | 1.348(9) | 1.348 | 0.0007 |
| C10–C11 | 1.478(7) | 1.478 | 0.0002 | C27–C28 | 1.338(10) | 1.338 | 0.0013 |
| C11–O1 | 1.184(6) | 1.184 | $1 \times 10^{-4}$ | C27–Cl1 | 1.719(6) | 1.719 | 0.0006 |
| C11–O2 | 1.370(6) | 1.37 | 0.0002 | C28–C29 | 1.388(8) | 1.388 | 0.0012 |
| C12–O2 | 1.439(6) | 1.439 | $1 \times 10^{-4}$ | N1–N2 | 1.374(5) | 1.374 | 0.0012 |
| C12–O3 | 1.375(5) | 1.375 | 0.0003 | N3–N4 | 1.415(5) | 1.415 | 0 |
| C13–C14 | 1.514(7) | 1.514 | 0.0008 | | | | 0.00046 |

**Table 3.** Comparative analysis of selected bond angles (°) for the investigated compound as obtained from SCXRD and DFT methods.

| A1–A2–A3 | Angle/° | | | A1–A2–A3 | Angle/° | | |
|---|---|---|---|---|---|---|---|
| | SCXRD | DFT | \|SCXRD-DFT\| | | SCXRD | DFT | \|SCXRD-DFT\| |
| C2–C1–C6 | 119.6(5) | 119.6 | 0.0075 | C14–C15–C22 | 112.9(6) | 112.9 | 0.053 |
| C3–C2–C1 | 121.2(6) | 121.2 | 0.0021 | C22–C15–C16 | 111.8(5) | 111.8 | 0.0004 |
| C2–C3–C4 | 119.0(6) | 119 | 0.0086 | C17–C16–C15 | 111.3(5) | 111.3 | 0.0005 |
| C5–C4–C3 | 121.3(6) | 121.3 | 0.0019 | C16–C17–C18 | 112.1(5) | 112.1 | 0.0158 |
| C4–C5–C6 | 120.0(5) | 120 | 0.0218 | C13–C18–C17 | 107.9(4) | 107.9 | 0.0135 |
| C1–C6–C7 | 120.8(4) | 120.8 | 0.0278 | C13–C18–C19 | 113.6(4) | 113.6 | 0.0003 |
| C5–C6–C1 | 118.9(5) | 118.9 | 0.0247 | C17–C18–C19 | 114.7(4) | 114.7 | 0.0156 |
| C5–C6–C7 | 120.2(4) | 120.2 | 0.0015 | C20–C19–C18 | 114.1(5) | 114.1 | 0.0354 |
| N1–C7–C6 | 123.7(4) | 123.7 | 0.0039 | C20–C19–C21 | 111.8(5) | 111.8 | 0.0051 |
| N1–C7–N3 | 110.6(5) | 110.6 | 0.0241 | C21–C19–C18 | 112.1(5) | 112.1 | 1.9646 |
| N3–C7–C6 | 125.7(4) | 125.7 | 0.0272 | N4–C23–C24 | 120.4(5) | 120.4 | 0.0036 |
| N2–C8–N3 | 109.9(4) | 109.9 | 0.0531 | C25–C24–C23 | 121.7(5) | 121.7 | 0.0051 |
| N2–C8–S1 | 126.0(4) | 126 | 0.0063 | C25–C24–C29 | 119.3(5) | 119.3 | 0.0219 |
| N3–C8–S1 | 124.0(4) | 124 | 0.0223 | C29–C24–C23 | 119.0(6) | 119 | 0.0345 |
| C10–C9–C12 | 108.6(4) | 108.6 | 0.0641 | C24–C25–C26 | 120.2(6) | 120.2 | 0.052 |
| C10–C9–S1 | 137.2(4) | 137.2 | 0.0181 | C27–C26–C25 | 120.7(7) | 120.7 | 0.9739 |
| C12–C9–S1 | 114.2(3) | 114.2 | 0.0544 | C26–C27–Cl1 | 120.5(6) | 120.5 | 0.0061 |
| C9–C10–Br1 | 132.2(4) | 132.2 | 0.0043 | C28–C27–C26 | 118.6(6) | 118.6 | 0.0434 |
| C9–C10–C11 | 109.4(4) | 109.4 | 0.0352 | C28–C27–Cl1 | 120.9(5) | 120.9 | 0.0369 |
| C11–C10–Br1 | 118.3(4) | 118.3 | 0.0264 | C27–C28–C29 | 121.6(7) | 121.6 | 0.0151 |
| O1–C11–C10 | 130.1(5) | 130.1 | 0.0156 | C24–C29–C28 | 119.4(7) | 119.4 | 0.0173 |
| O1–C11–O2 | 122.4(5) | 122.4 | 0.0207 | C7–N1–N2 | 107.8(4) | 107.8 | 0.0185 |
| O2–C11–C10 | 107.5(5) | 107.5 | 0.0274 | C8–N2–N1 | 106.8(4) | 106.8 | 0.0146 |
| O2–C12–C9 | 104.2(3) | 104.2 | 0.0415 | C7–N3–C8 | 104.9(4) | 104.9 | 0.0364 |
| O3–C12–C9 | 109.2(4) | 109.2 | 0.0538 | C7–N3–N4 | 127.6(4) | 127.6 | 0.0446 |
| O3–C12–O2 | 111.3(4) | 111.3 | 0.0239 | C8–N3–N4 | 125.7(4) | 125.7 | 0.0351 |
| C18–C13–C14 | 111.9(4) | 111.9 | 0.0428 | C23–N4–N3 | 115.1(4) | 115.1 | 0.043 |
| O3–C13–C14 | 111.7(4) | 111.7 | 0.0257 | C11–O2–C12 | 110.2(4) | 110.2 | 0.0102 |
| O3–C13–C18 | 106.9(4) | 106.9 | 0.0711 | C12–O3–C13 | 115.7(3) | 115.7 | 0.0212 |
| C15–C14–C13 | 111.6(5) | 111.6 | 0.0034 | C9–S1–C8 | 103.3(2) | 103.3 | 0.0408 |
| C14–C15–C16 | 108.7(5) | 108.7 | 0.0249 | | | | 0.071549 |

## 2.2. Hirshfeld Surface Analysis

The Hirshfeld surface analysis [18] for the title compound was performed using a sequence of steps. The analysis was accomplished with the aid of the Crystal Explorer program [5]. Initially, the two-dimensional fingerprint plots were calculated for the compound's crystal. This step also included the calculation of the electrostatic potentials [7]. In the next step, these electrostatic potentials were mapped onto the Hirshfeld surfaces. This mapping was conducted using the 6-31G(d,p) basis set at the level of Density Functional Theory (DFT) [12].

The crystallographic information file (CIF) of the compound was utilized as the input for the analysis [1]. It is noteworthy that the normalization of the bond lengths of hydrogen atoms involved in interactions was crucial for the generation of the fingerprint plots. The bond lengths were normalized to standard neutron values, where C–H = 1.083 Å, N–H = 1.009 Å, and O–H = 0.983 Å [12].

### 2.3. Energy Framework Study

The energy framework study is a powerful method used to understand the topology of the overall interactions of molecules within a crystal [18]. This technique enables the calculation and comparison of various energy components, such as repulsion ($E_{rep}$), electric ($E_{ele}$), dispersion ($E_{dis}$), polarization ($E_{pol}$), and total ($E_{tot}$) energy. The energy calculations are based on the anisotropy of the topology of pairwise intermolecular interaction energies.

In the study of the title compound, the Crystal Explorer program [5] was employed to determine the energy framework. The computational procedure involved generating new wave functions using the Density Functional Theory (DFT) method with the 6-31G(d,p) basis set. This was performed with the incorporation of exchange and potential functions (B3LYP) configured for a molecular cluster environment within a $1 \times 1 \times 1$ unit cell.

The energy framework representation includes cylinders, whose thickness is indicative of the intensity of interactions. This thickness is directly proportional to the energy magnitude, providing valuable insights into the stabilization of the crystal packing [12].

### 2.4. Computational Details

The computational detail study was conducted using a series of steps, beginning with Density Functional Theory (DFT) calculations. These calculations were performed using the experimental single-crystal X-ray data from previous work [12,19,20] as input geometries.

The geometry optimizations of different monomer models were accomplished in the gas phase using the Gaussian 09, Rev D.01 software package [21]. For visualization, analysis, modification, and exportation of data results, the GaussView 6.0 program [22] was employed. The wB97XD functional, a version of Grimme's D2 dispersion model developed by Head-Gordon et al. [23], was utilized in this study. This functional includes long-range corrections and has been found to be most effective for describing hydrogen bonding interactions [24,25]. Frequency calculations on the optimized geometry confirmed that all stationary points were true minima (zero imaginary frequency) on the potential energy surface.

All global and local chemical reactivity descriptors for the systems were calculated using the concepts of Conceptual DFT (CDFT), also known as Chemical Reactivity Theory (CRT) [26–29]. The electrophilic $P_k^+$ and nucleophilic $P_k^-$ Parr functions [20,30] were determined through the analysis of the Mulliken Atomic Spin Density (ASD) of the radical anion and the radical cation starting from optimized neutral geometries. For single-point energy calculations, an unrestricted open-shell DFT scheme was used, specifically UwB97XD/6.311G(d,p). Charge and multiplicity were defined as (+1, 2) for cations and (−1, 2) for anions, respectively, using the MULTIWFN program [31]. The GaussSum 3.0 software was employed to compute the contributions of molecular orbitals associated with specific functional groups within the molecule [31,32]. Additionally, the program facilitated the generation of both the Density of States (DOS) spectrum and the Partial Density of States (PDOS) spectrum. Finally, using the MULTIWFN program, various aromaticity indices and LOLIPOP amounts were determined for the title compounds [6].

## 3. Results

### 3.1. A Crystallographic Study of Molecular Geometries and Supramolecular Characteristics

The molecular structure of (*E*)-3-bromo-4-((4-((1-(4-chlorophenyl)ethylidene)amino)-5-phenyl-4H-1,2,4-triazol-3-yl)thio)-5-((2-isopropylcyclohexyl)oxy)furan-2(5H)-one features an array of distinct moieties and functional groups Figure 1. Central to its architecture is an ethylidene bridge connecting a 4-chlorophenyl ring to a 1,2,4-triazole ring. This triazole

ring has a phenyl group at its fifth position, lending aromatic character to that region of the molecule. Additionally, the triazole and furanone rings are connected by a thioether linkage. The furanone ring, an oxygen-containing five-membered heterocycle, bears a bromine atom at its third position and has an oxy-linked isopropyl-substituted cyclohexyl ring at the fifth position. It is noteworthy to mention that this compound is cataloged under the CCDC code 829447 [1] and has an empirical formula of $C_{29}H_{30}BrClN_4O_3S$, weighing 629.99 g/mol. Its crystallographic analysis was conducted at a temperature of 293(2) K, revealing a monoclinic crystal system in the C2 space group. Key geometric parameters include unit cell dimensions of a = 33.795(5) Å, b = 8.871(5) Å, and c = 10.039(5) Å, with interaxial angles of $\alpha$ = 90.000(5)°, $\beta$ = 98.337(5)°, and $\gamma$ = 90.000(5)°. The crystal has a volume of 2978(2) Å$^3$, a calculated density of 1.405 g/cm$^3$, and was examined using MoK$\alpha$ radiation. The refined structural parameters include R1 = 0.0499 for reflections with I $\geq$ 2$\sigma$(I) and a goodness-of-fit on F$^2$ of 0.907.

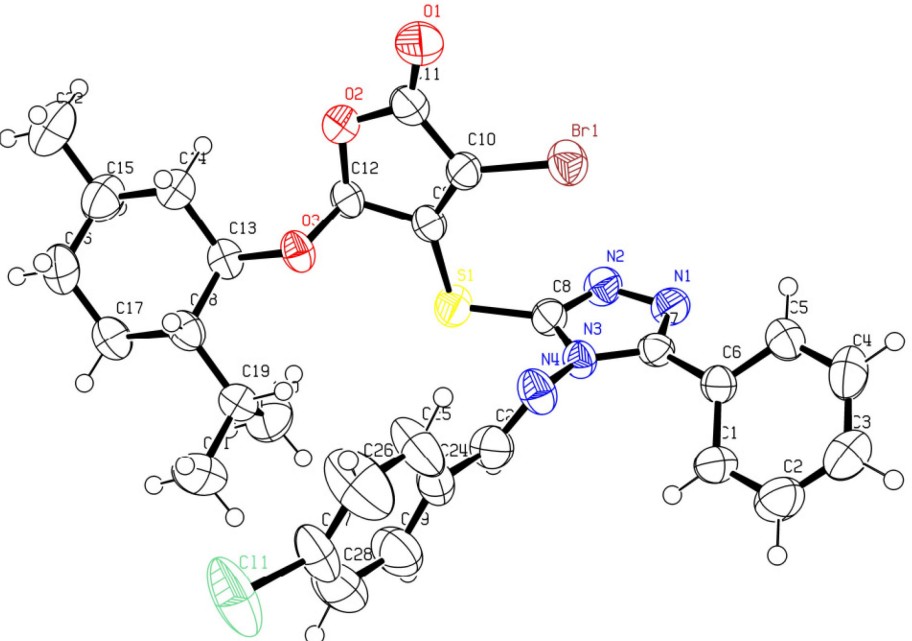

**Figure 1.** The non-symmetrical element of the subject molecule, along with its atomic index. Displacement ellipsoids, illustrated at a 50% likelihood level, are also included.

Table 1, Figures 2 and 3A,B provide comprehensive data elucidating the nature of the intermolecular interactions in the system under consideration. For the O1$\cdots$H1 interaction, the observed bond length is 2.454 Å, which is shorter by 0.266 Å than the expected combined van der Waals (VdW) radii. This deviation, accompanied by an Rv value of 9.78, indicates the presence of a potent hydrogen bond. Similarly, the H16B$\cdots$Cl1 interaction, with a bond length of 2.936 Å and a minor deviation of 0.014 Å from the combined VdW radii, suggests van der Waals interactions that might be augmented with dipole–dipole interactions, as evidenced by its Rv value of 0.47.

The C10$\cdots$H5 interaction, which exhibits a bond length of 2.881 Å and deviates from the combined VdW expectation by 0.019 Å, can be interpreted as a mild van der Waals or dispersion interaction, supported by its Rv value of 0.66. Contrarily, the C12$\cdots$N1 interaction shows a bond length of 3.184 Å, which is less by 0.066 Å than the combined VdW radii. Its Rv value of 2.03, however, suggests a relatively weak interaction.

Lastly, the H12$\cdots$N1 interaction, characterized by a bond length of 2.43 Å and a notable deviation of 0.32 Å from the expected VdW radii, is indicative of a strong hydrogen bond. This assertion is further reinforced by the substantial Rv value of 11.64, corroborating the strength and nature of the interaction.

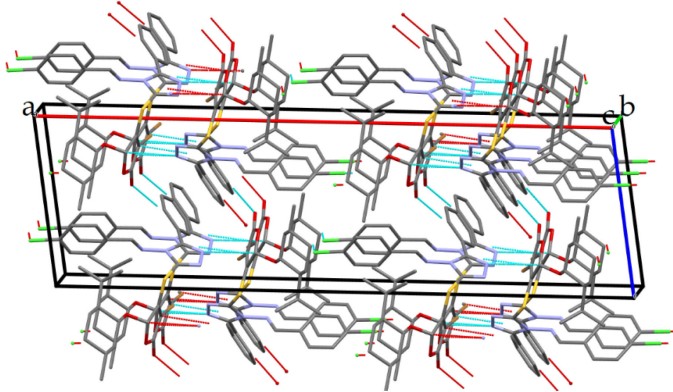

**Figure 2.** A partial packing diagram of the tested compound, viewed from the b-axis orientation. The diagram emphasizes short-distance contact environments, shorter than the sum of the van der Waals radii. Non-participating hydrogen atoms in hydrogen bonding have been omitted for improved clarity and understanding.

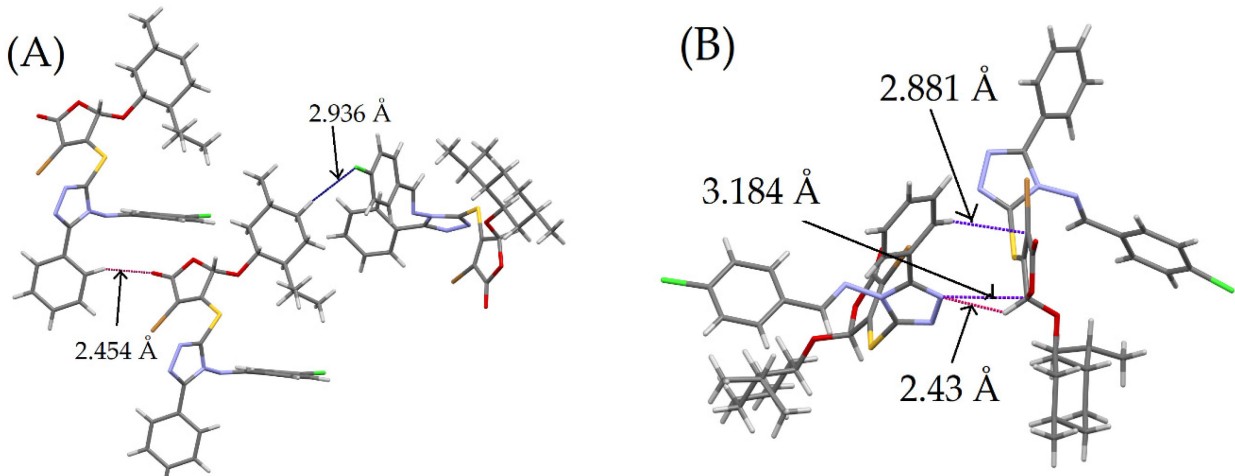

**Figure 3.** Partial packing diagram (**A**) trimer models, showing the C1–H1· · ·O1 (red dashed lines) and C–H16B· · ·Cl1 van der Waals interactions with potential dipole–dipole interactions (blue dashed lines) and (**B**) dimer models C12–H12· · ·N1 (red dashed lines), C12· · ·N1 (blue dashed lines), and C5–H5· · ·C10 interactions (blue dashed lines).

The geometric characteristics obtained from X-ray diffraction (SCXRD) were compared with those obtained from Density Functional Theory (DFT) computations. The mean absolute errors (MAEs) for bond lengths are reported in Table 2, indicating an MAE of less than 0.00046 Å. In the given context, where the average bond length is around 1.447 Å, the mean absolute error (MAE) obtained substantiates the effectiveness of the wB97XD functional in accurately predicting the bond lengths for the specific compound under consideration. In a similar vein, the mean absolute error (MAE) for bond angles is found to be below 0.071549°. When compared to the standard bond angle of 116.361°, as presented in Table 3, this further emphasizes the effectiveness of the wB97XD functional. Collectively, these assessments provide evidence of the wB97XD functional's ability to accurately and precisely determine the geometric properties of the compound under investigation.

### 3.2. Hirshfeld and Other Surfaces

The Hirshfeld surface analysis is a powerful tool for identifying and analyzing intermolecular interactions in the crystal packing of a compound. In the given context, the analysis was performed for a certain test compound, with the normalized contact distance ($d_{norm}$) based on internal ($d_i$) and external ($d_e$) distances covering the range from

0.1479 (red) to 1.6853 (blue). From the 3D Hirshfeld surface maps, regions with an intense red color on the dnorm are located over the oxygen (O1), nitrogen (N1), and hydrogen (H1 and H12) atoms of the tested compound. The "red spots" denote the presence of strong hydrogen bonds, particularly the intermolecular interactions of O1···H1 (2.454 Å) and C12···N1 (2.430 Å), as detailed in Table 1 and Figure 4.

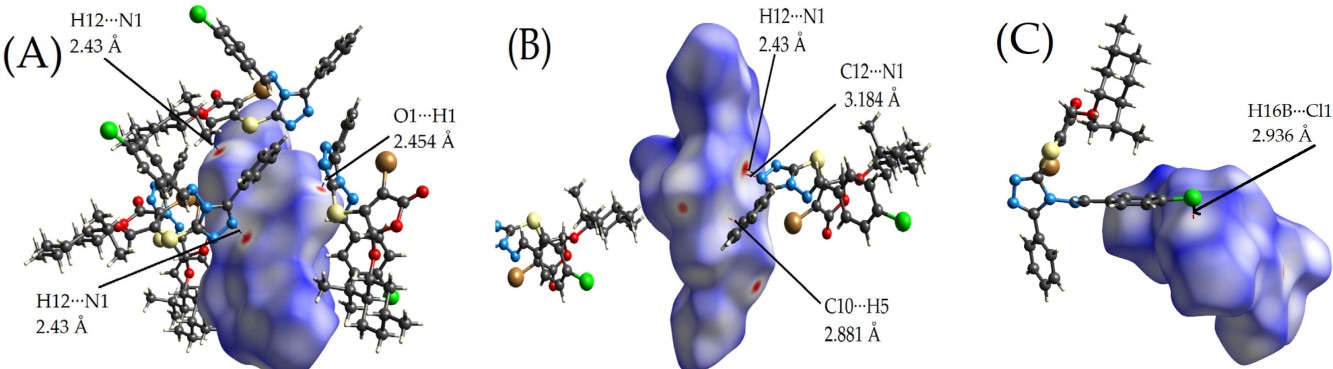

**Figure 4.** The graphical interface of CrystalExplorer21, showcasing a Hirshfeld surface (HS) that envelops the molecules in their crystalline assembly (CCDC code 829447), juxtaposed with neighboring molecules. Panel (**A**) distinctly illustrates the prominent intermolecular interactions of O1···H1 (2.454 Å) and C12···N1 (2.430 Å). These hydrogen bonds, which link an atom from the surface to exterior hydrogen atoms or those on its antipode, are identifiable. They traverse the epicenter of the red-shaded dnorm region. On the other hand, panels (**B**,**C**) depict weaker interactions. These are associated with the surface but do not course through the heart of the red-hued dnorm area. Specifically, panel (**B**) captures the van der Waals interaction observed in the C12···N1 contact (3.184 Å) and the C10···H5 contact m(2.881 Å), while panel (**C**) displays the van der Waals interaction associated with the H16B···Cl1 contact (2.936 Å).

In addition to the Hirshfeld surface (Figure 5A,B), the Shape Index and Curvedness surfaces are integral in discerning distinct molecular packing modalities, specifically $\pi$··· $\pi$ stacking.

The Shape Index, which extends the molecule (signifying a concave shape) to 1 (signifying a convex shape), serves as a reflection of the electron density surface shape, thereby revealing the nature of molecular interactions through $\pi$··· $\pi$ stacking. The Shape Index identifies $\pi$··· $\pi$ stacking via the juxtaposition of red and blue triangles; an absence of these adjacent triangles implies a lack of $\pi$··· $\pi$ interactions. This inference is supported by Figure 4B, which unequivocally indicates an absence of $\pi$··· $\pi$ interactions in the molecule under consideration due to the unavailability of adjacent red and blue triangles.

This conclusion can be further substantiated by an analysis of the Curvedness surface. The Curvedness surface, delineated from −4 to 4 Å, quantifies the degree of curvature exhibited by the comprehensive Hirshfeld map. As per Figure 5C, the Curvedness surface of the compound under study demonstrates the presence of numerous small green flat regions demarcated by blue edges, indicative of low Curvedness values. These regions provide evidence of a lack of flat surface patches over both sides of the molecular rings, thereby corroborating the absence of $\pi$··· $\pi$ stacking interactions with adjacent molecules.

The obtained results from the Hirshfeld surface analysis for the tested compound show that the $d_e$ values range from 1.0394 to 2.8558 Å while the $d_i$ values range from 1.0392 to 2.9445 Å. The close range of $d_e$ and $d_i$ values suggests that there is a balance between internal and external molecular interactions in the crystal structure of the tested compound. The small values for both $d_e$ and $d_i$ imply that the molecules in the crystal structure are closely packed.

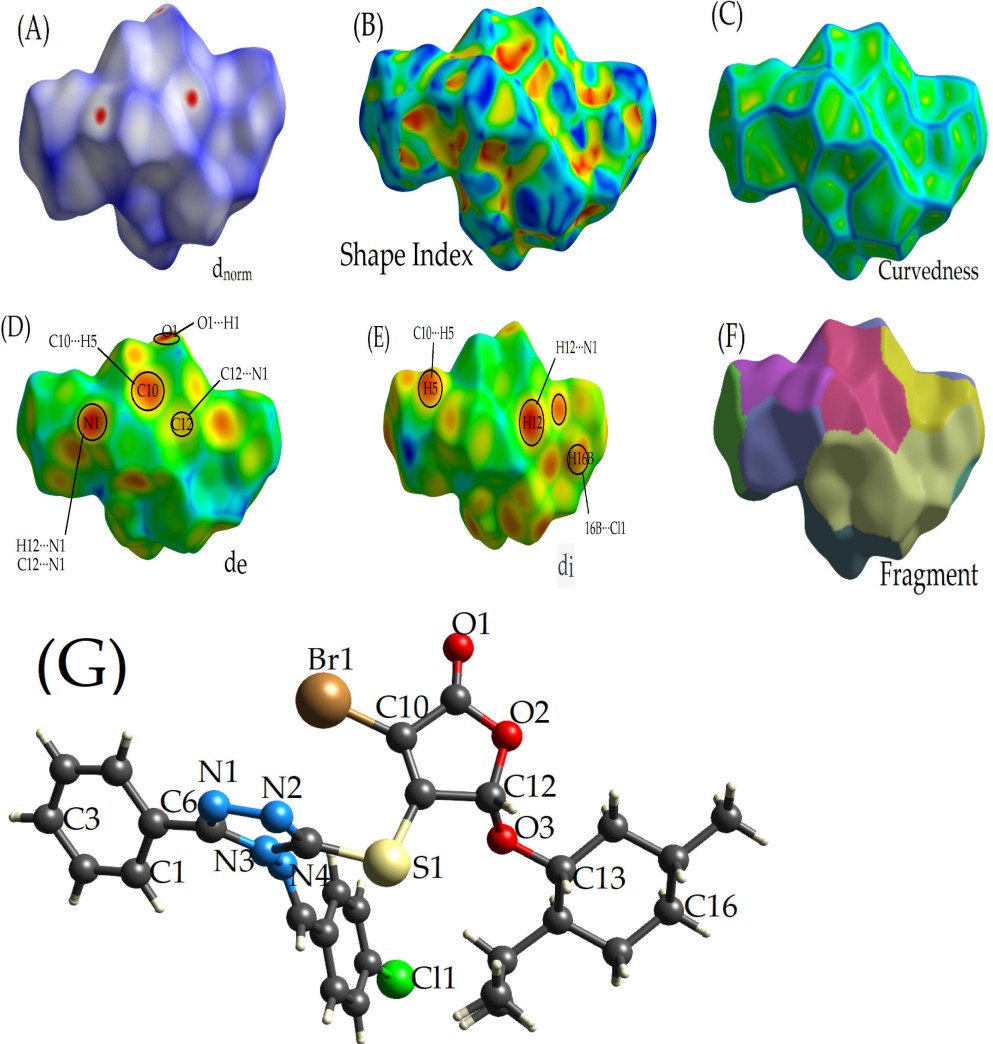

**Figure 5.** Depiction of the Hirshfeld surface for the investigated compounds with mappings for
(**A**) the normalized contact distance ($d_{norm}$), (**B**) Shape Index, (**C**) Curvedness, (**D**) electron density
exterior to the surface ($d_e$), (**E**) electron density interior to the surface ($d_i$), and (**F**) Fragment Patch
surfaces. The labeled structure for the compound under study is also included (**G**).

In Figure 5D,E, the $d_e$ and $d_i$ surfaces are depicted. To enhance the clarity between the
two conformations, the color gradient was consistently set: red corresponds to a value of
1.0629 a.u and blue to 2.6497 a.u.

The $d_e$ values detail the distance to the closest external atom, spanning a range from
1.0394 Å to 2.8558 Å, as presented in Figure 5D. Conversely, the $d_i$ values, indicative of the
distance to the proximate internal atom, fluctuate between 1.0392 Å and 2.9445 Å, as shown
in Figure 5E. Lower $d_i$ values insinuate that molecular atoms are proximal to the Hirshfeld
surface, whereas elevated values could signify regions where atoms reside more internally.
Reduced $d_e$ values may allude to potent intermolecular interactions with neighboring
molecules, attributable to forces like hydrogen bonding or π–π stacking. Elevated $d_e$ values
might indicate molecular regions with minimal proximate interactions, potentially resulting
from steric impediments or the lack of congruent intermolecular forces.

In the examined compound, H1 and H12 emerge as the immediate external nuclei
from the $d_e$ surface's perspective. Interestingly, these identical hydrogen atoms also present
as the closest internal donor nuclei relative to $d_i$, manifesting as discernible red regions
within the contour plots. Conversely, O1 and N1 are pinpointed as the immediate external
nuclei from the $d_i$ surface's viewpoint.

This alignment underscores the prominence of robust hydrogen bond intermolecular interactions, namely C1–H1···O1 and C12–H12···N1, within the title compound. This observation is in harmony with the Hirshfeld surface patterns superimposed on electrostatic potentials and $d_{norm}$. Additional red regions on these surfaces pertain to short-range interactions, detailed further in Figure 5E and Table 1. Moreover, the notable absence of green flat terrains on both surfaces corroborates the nonexistence of π···π stacking interactions within the crystal's architecture.

Finally, the Fragment Patch surfaces, mapped between 0 and 14 Å for the tested compound, help identify the closest neighbor coordination environment of a molecule (Figure 5F). This mapping of color patches provides an additional perspective on the compound's intermolecular interactions and overall crystal packing.

The results provided represent the percentage contributions of different types of intermolecular interactions to the total Hirshfeld surface for the tested compound. These are derived from a method known as "fingerprinting" intermolecular interactions in molecular crystals, which provides a detailed and quantitative view of the types and proportions of different intermolecular contacts [7]. From the data in Figures 6 and 7, the most significant contribution to the intermolecular interactions is the H···H contact, which makes up 39.1% of the interactions. These are likely to be van der Waals interactions, which are often the dominant interactions in molecular crystals [6]. The N···H/H···N and O···H/H···O contacts, which contribute 8.8% and 11%, respectively, likely represent hydrogen bonding interactions. Weak bonds involving nitrogen and oxygen atoms are common and can significantly influence the crystal packing [33]. The C···H/H···C contacts, accounting for 15.9%, are likely to be CH–π interactions, especially if aromatic rings are present in the structure, which agreed with Shape Index result [34]. The interactions involving halogens (Br and Cl), such as Br···H/H···Br and Cl···H/H···Cl, which contribute 6.6% and 8.6%, respectively, could be indicative of a van der Waals interaction with potential dipole–dipole interactions [35]. The other contacts listed have lower contributions, suggesting they have less influence on the overall crystal packing of the tested compound. However, they could still play a role in determining the specific details of the crystal structure [5].

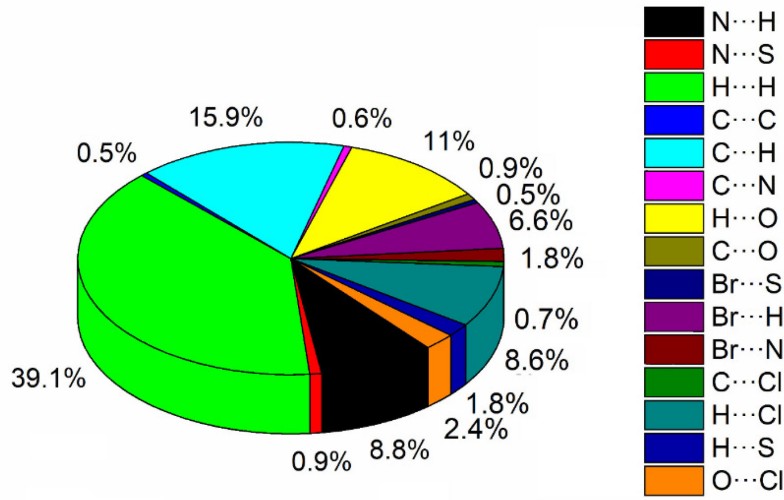

**Figure 6.** Representation of various intermolecular interactions as determined by Hirshfeld surface analysis.

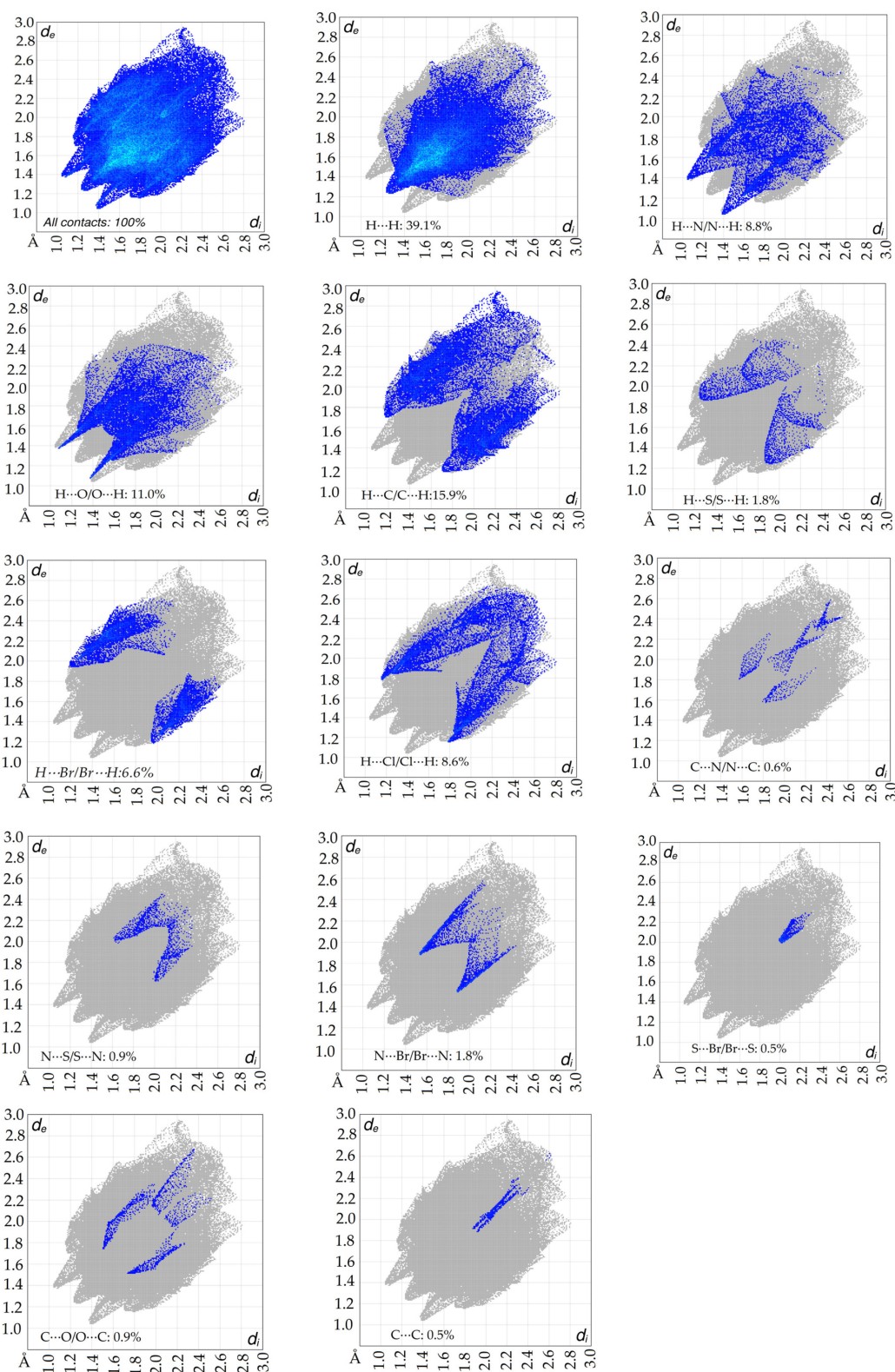

**Figure 7.** Fingerprint plots of Hirshfeld surface analysis showing the correlation between the nearest internal ($d_i$) and external ($d_e$) distances for the tested compound. The color gradation, from light to dark blue, indicates the frequency of points with identical $d_i$, $d_e$ coordinates (light blue corresponds to higher frequencies, dark blue to lower frequencies).

*3.3. Energy Frameworks*

The energy framework serves as a critical tool for understanding the distinct energy types that contribute to the supramolecular assembly of molecules within a crystal [18]. In this study, the energy framework computations were performed using the CrystalExplorer 21.5 software [5], a robust tool recognized for this type of analysis. These calculations were based on a B3LYP/6-31G(d,p) functional basis set, a widely accepted basis set in computational chemistry. The interaction energies were calculated for a 3.8 Å cluster around a single molecule of the tested compound. The chosen scale factors are consistent with previously established values [18,36,37]. The three-dimensional energy frameworks of the title chemical are illustrated in Figure 8. This figure emphasizes the neighboring molecules located within a radius of 3.8 Å from the reference molecule. The reference molecule is represented by a black ball and stick model. The frameworks are depicted using cylindrical shapes that are assigned different colors, namely red, green, and blue, to symbolize the Coulombic (classical electrostatic, $E_{ele}$), dispersion ($E_{dis}$), and total energy ($E_{tot}$) components, respectively [38–40]. The visualization depicted in Figure 9 encompasses the crystallographic axes a, b, and c.

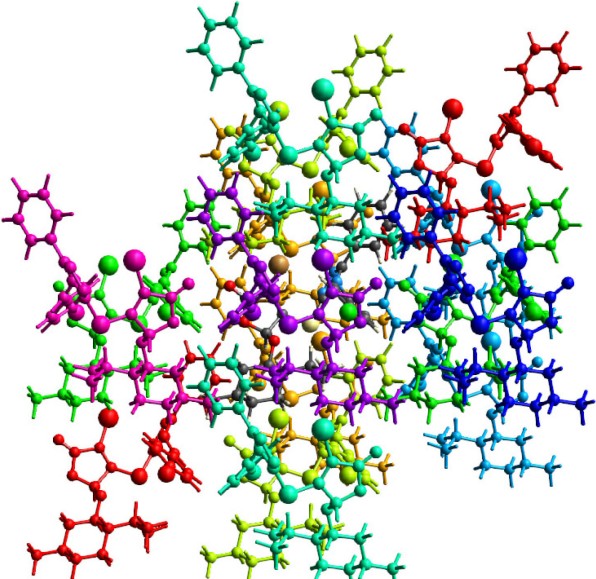

**Figure 8.** Depiction of the three-dimensional energy frameworks, showcasing the molecules adjacent to the reference molecule (represented in black ball and stick model) within a default radius of 3.8 Å. The surrounding molecules are color-coded for differentiation and displayed in multiple orientations.

The varying cylinder thicknesses across the energy frameworks signify the relative interaction strengths between the molecules, as further validated by the notably high negative energy values presented in Table 4. This table not only provides an energy breakdown but also offers details such as the color-coding scheme correlating to molecular interactions at specific Cartesian coordinates, the number of interactions involving the central molecule (N), the distance between molecular centroids (R), and the associated rotational symmetry operations (symmetry). Such details are pivotal for lattice energy calculations, as mentioned in references [41–43].

The presented data in Table 4 for the energy frameworks depict the energy contributions of different components, namely $E_{ele}$ (electrostatic), $E_{pol}$ (polarization), $E_{dis}$ (dispersion), $E_{rep}$ (repulsion), and $E_{tot}$ (total) for various intermolecular interactions in the compound.

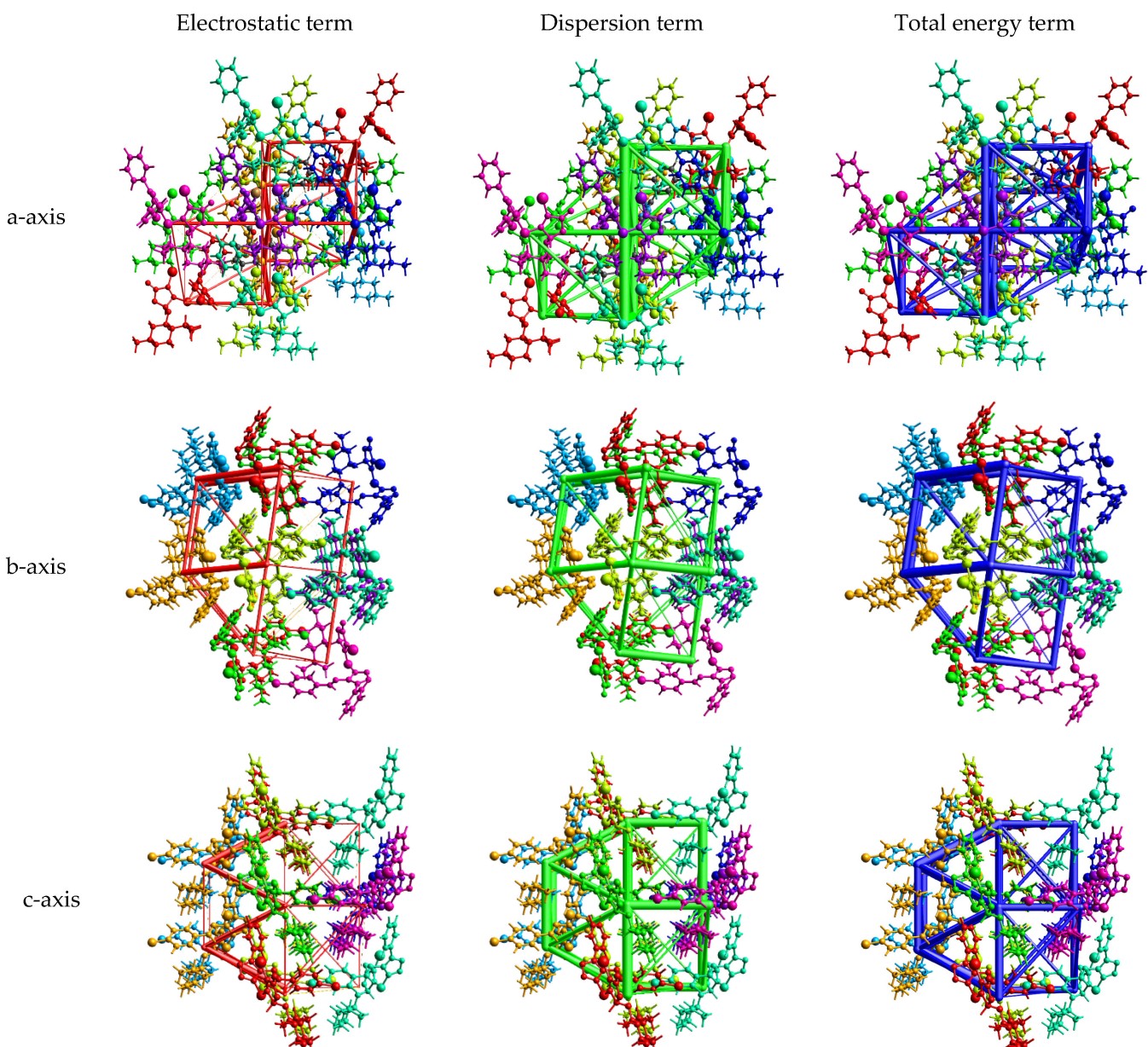

**Figure 9.** Various energy structures for the compound, comprising electrostatic energy, dispersion energy, and cumulative energy elements.

The analysis reveals a total interaction energy ($E_{tot}$ = −51.66 kJ/mol) linked with the C12–H12···N1 interaction, associated with a symmetric pair of orange molecules distanced at R = 9.02 Å. Similarly, the C1–H1···O1 interaction, with an energy of $E_{tot}$ = −37.84 kJ/mol, corresponds to a symmetric pair of pale-green molecules situated R = 10.04 Å apart. It is essential to note that these interaction energies, derived from energy frameworks, encompass the overall molecular interactions and are not exclusively representative of individual contacts.

**Table 4.** Different interaction energies of the molecular pairs in kJ/mol. Electron density was B3LYP/6-31G(d,p).

| No. | | N | Symop | R | Electron Density | Eele | Epol | Edis | Erep | Etot |
|---|---|---|---|---|---|---|---|---|---|---|
| 1 | | 2 | x, y, z | 13.40 | B3LYP/6-31G(d,p) | 0.9973 | −1.1749 | −23.3781 | 0.0000 | −20.1725 |
| 2 | | 2 | −x + 1/2, y + 1/2, −z | 9.02 | B3LYP/6-31G(d,p) | −30.5215 | −9.6167 | −43.1703 | 40.9923 | −51.6576 |
| 3 | | 2 | x, y, z | 8.87 | B3LYP/6-31G(d,p) | −5.7677 | −4.9333 | −59.4765 | 30.9113 | −42.4465 |
| 4 | | 2 | x, y, z | 10.04 | B3LYP/6-31G(d,p) | −17.2125 | −7.1891 | −32.3614 | 22.4292 | −37.8438 |
| 5 | | 2 | −x, y, −z | 12.73 | B3LYP/6-31G(d,p) | −7.8179 | −0.4275 | −13.4144 | 0.0000 | −20.2634 |
| 6 | | 2 | −x + 1/2, y + 1/2, −z | 11.94 | B3LYP/6-31G(d,p) | −11.5618 | −2.5711 | −21.0371 | 9.9290 | −26.3126 |
| 7 | | 1 | −x, y, −z | 13.92 | B3LYP/6-31G(d,p) | 3.0228 | −0.6213 | −11.2461 | 0.0000 | −7.0568 |
| 8 | | 1 | −x, y, −z | 9.12 | B3LYP/6-31G(d,p) | −5.7749 | −1.3866 | −55.3823 | 29.4028 | −37.1966 |
| 9 | | 1 | −x, y, −z | 13.21 | B3LYP/6-31G(d,p) | 0.3536 | −0.1993 | −9.5310 | 0.0000 | −8.0732 |

For the $C16–H16B \cdots Cl1$ interaction, the delineation of molecular interaction energy components is intricate. $E_{ele}$ (−7.8179 kJ/mol) highlights dominant electrostatic attractions, $E_{pol}$ (−0.4275 kJ/mol) indicates electron cloud reorientations suggesting charge redistribution, and $E_{dis}$ (−13.4144 kJ/mol) points to the presence of significant van der Waals interactions. $E_{rep}$ being zero indicates an absence of repulsive electron cloud overlap. The cumulative energy $E_{tot}$ (−20.2634 kJ/mol) suggests a dominant interaction, predominantly driven by electrostatic and dispersion forces, considering the total molecular interaction. The absence of repulsion might point towards an optimal molecular arrangement or balancing forces at play. The nature of the bond seems to be primarily of van der Waals type with potential dipole–dipole interactions, based on the comprehensive interaction of the molecules. However, a solely covalent bond appears less probable in this scenario.

Lastly, the interaction energy with the smallest magnitude is $E_{tot}$ = −7.06 kJ/mol, corresponding to a blue molecular pair separated by the most extended centroid distance of R = 13.92 Å. This again is a representation of the complete interaction and not just specific contacts.

In the context of energy conversion factors ($k_{ele}$, $k_{dis}$, $k_{pol}$, and $k_{rep}$), values are provided for two benchmarked energy models: CE-HF...HF/3-21G and CE-B3LYP...B3LYP/6-31G(d,p). These are appropriately scaled and presented in the tables below [44]. Overall, this energy framework analysis underscores the dominance of the $E_{dis}$ component over the classical electrostatic ($E_{ele}$) component.

In the presented data (Table 4), electrostatic interactions, characterized by charge differences between entities, show the $C12–H12 \cdots N1$ interaction as having the most pronounced negative $E_{ele}$, indicative of a potential strong hydrogen bond. Dispersion interactions, resulting from electron density fluctuations, manifest significantly in interactions like $H25 \cdots H20B$ and $H26A \cdots H26A$, pointing to the involvement of non-polar regions. Polarization interactions, where one molecule's electron cloud is influenced by another's electric field, are prominent in the $C12–H12 \cdots N1$ and $C1–H1 \cdots O1$ interactions, suggesting the presence of polar groups. The positive $E_{rep}$ values in certain interactions denote repulsive forces, which are typical when atomic proximities surpass van der Waals limits. Cumulatively, $E_{tot}$ provides an integrated view of interaction strength, with the $C12–H12 \cdots N1$ interaction exhibiting the most pronounced negative energy, reflecting substantial electrostatic and dispersion influences.

A closer look at the calculated energies reveals significant insights. The electrostatic, polarization, dispersion, and repulsion energies were calculated to be −74.28 kJ/mol, −28.12 kJ/mol, −268.997 kJ/mol, and 133.66 kJ/mol, respectively. The total energy was found to be −251.023 kJ/mol. This total energy is a crucial determinant of the stability of the crystal structure.

Most notably, the dispersion interaction energy dominates amongst the other interaction energies. This suggests that van der Waals forces, which contribute to dispersion interactions, play a significant role in the supramolecular assembly of the compound in the crystal structure. Such forces drive the formation of the crystal lattice, and this information could be invaluable in the design and synthesis of similar crystalline materials.

*3.4. Chemical Reactivity Properties*

3.4.1. Analysis of CDFT Reactivity Indices

a.    Frontier molecular orbitals (FMOs) for the investigated compound

Frontier molecular orbitals (FMOs), specifically the Highest Occupied Molecular Orbital (HOMO) and the Lowest Unoccupied Molecular Orbital (LUMO), are pivotal descriptors in computational quantum chemistry. The energy gap between these orbitals, often termed the HOMO-LUMO gap, provides insight into molecular stability, reactivity, and electronic properties [34]. A narrower gap often suggests enhanced electrical conduction and reduced kinetic stability. HOMO characterizes electron-donating ability, while LUMO represents electron-accepting capacity [35]. For the compound under study, out of its 668 molecular orbitals, 162 are occupied. The HOMO and LUMO are designated as the 162nd and 163rd orbitals, respectively, with energies of $-8.43$ eV and $-0.4$ eV.

The Density of States (DOS) spectrum offers insights into energy levels along a specified energy bandwidth (Figure 10), $\Delta E$. This spectrum, when convoluted with contributions from specific atomic or functional groups, is termed the Partial Density of States (PDOS) (see Figure 11) [36]. Through the data given, the electronic contributions of various molecular fragments to the Lowest Unoccupied Molecular Orbital (LUMO) and the Highest Occupied Molecular Orbital (HOMO) can be ascertained.

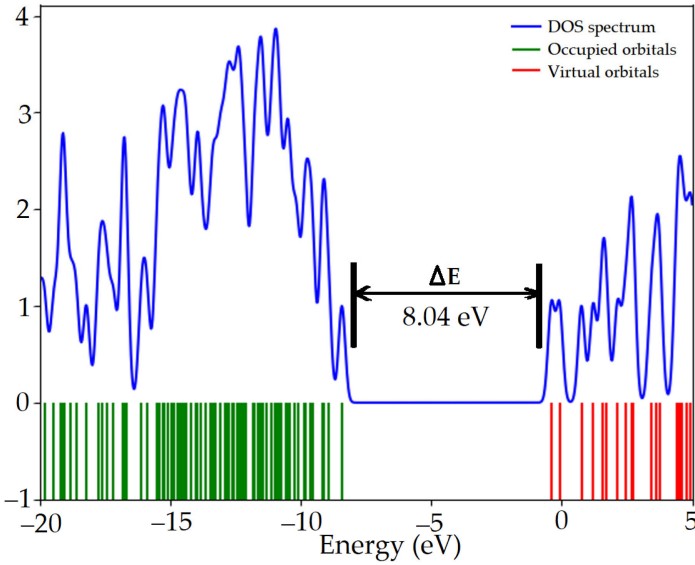

**Figure 10.** Computed total electronic Density of States for the investigated molecule.

As depicted in Figures 10, 11 and 12A, the HOMO, located at $-8.43$ eV and characterized by symmetry A, predominantly draws contributions from the phenyl $C_6H_5$ and 4H-1,2,4-triazol-4-amine ($C_3N_4$) fragments, which account for 38% and 35%, respectively. This suggests that these groups chiefly govern the occupied electronic states, underscoring their potential role as electron donors in chemical interactions. The 3-bromofuran-2,5-dione ($SBrC_5H_3O_2$) fragment also imparts a significant influence on the HOMO, contributing 25%. Notably, the methynimine N=CH, $C_6H_4Cl$, and 1-isopropyl-4-methylcyclohexane $C_{10}H_{19}$ fragments have negligible influence on the HOMO, with their contributions being 0%.

Conversely, as presented in Figures 10, 11 and 12B, the LUMO, positioned at $-0.4$ eV with symmetry A, is majorly influenced by the methynimine (N=CH) and p-chlorophenyl ($C_6H_4Cl$) groups, which account for 46% and 45%, respectively. These groups are inferred to largely contribute to the unoccupied electronic states, highlighting their probable significance in electron acceptance during molecular interactions. The 4H-1,2,4-triazol-4-amine ($C_3N_4$) and 3-bromofuran-2,5-dione ($SBrC_5H_3O_2$) fragments offer modest contributions to the LUMO, at 4% and 2%, respectively, while the phenyl ($C_6H_5$) and 1-

isopropyl-4-methylcyclohexane ($C_{10}H_{19}$) fragments have the least contributions of 2% and 1%, respectively.

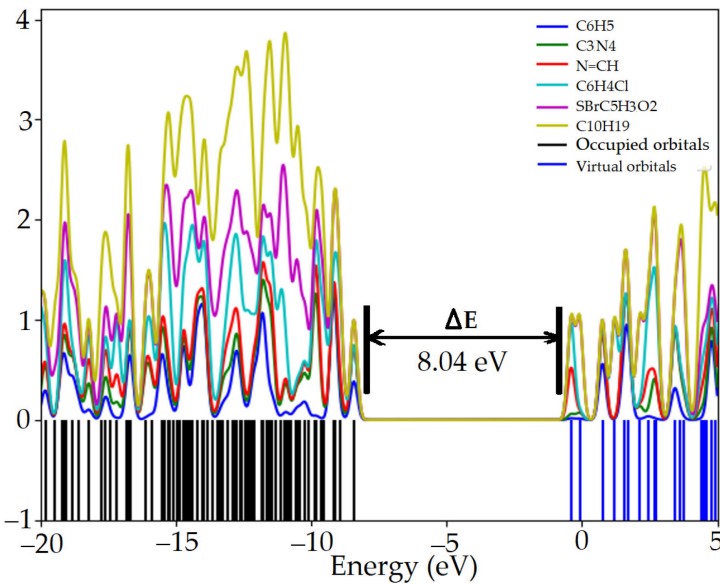

**Figure 11.** Computed Partial Density of Electronic States for the specified molecule.

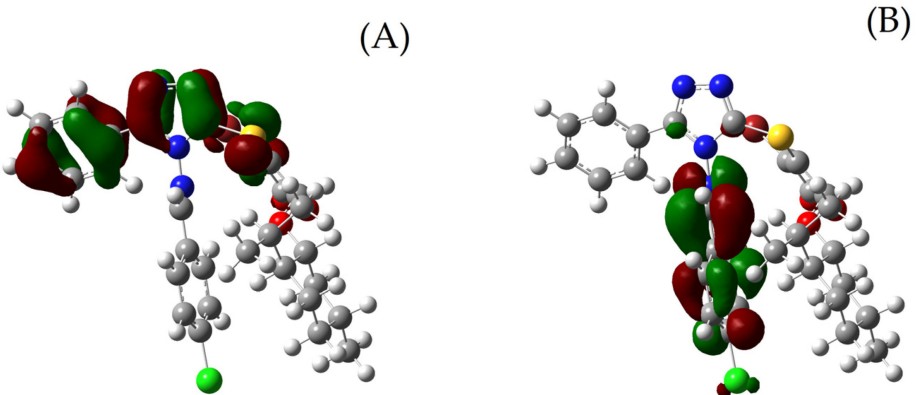

**Figure 12.** Representation of the frontier molecular orbitals of the designated molecule, computed via the DFT approach with the wB97XD functional and the 6-31G(d) basis set in the gas phase: (**A**) Highest Occupied Molecular Orbital (HOMO); (**B**) Lowest Unoccupied Molecular Orbital (LUMO).

b.　Global Reactivity Descriptors for the Investigated Compound

The Conceptual Density Functional Theory (CDFT), also known as Chemical Reactivity Theory (CRT), is a valuable approach that is becoming increasingly popular in organic chemistry. It can provide insightful information about the reactivity behavior of compounds in their ground states through the calculation of various descriptors, such as electronic chemical potential ($\mu$), chemical hardness ($\eta$), global electrophilicity ($\omega$), and global nucleophilicity (N) [45].

For the compound under investigation, these CDFT descriptors were calculated and are presented in Table 5 and Figure 9. The global electrophilicity index ($\omega$) for the tested compound was found to be 1.4134 eV. According to the electrophilicity scale, this value categorizes the compound as a moderate electrophile. The nucleophilicity index (N), another important descriptor, was calculated to be 0.8681 eV. As this value is less than 2.00 eV, the compound could be considered as a marginal nucleophile, suggesting that it may not readily donate electrons in a chemical reaction. The electronic chemical potential ($\mu$) of the compound was calculated to be $-4.5031$ eV. This value is important for understanding

the tendency and direction of electron density transfer during a chemical reaction involving the studied compound. A negative chemical potential suggests that the compound has a tendency to lose electrons, supporting its classification as a moderate electrophile [46].

**Table 5.** Values of global reactivity descriptors (in eV) for analyzed compounds, calculated using the wB97X-D/6.31G (d,p) level of theory.

| Vertical IP (eV) | Vertical EA (eV) | Mulliken Electronegativity (eV) | Chemical Potential (eV) | Hardness (eV) | Softness (eV$^{-1}$) | Electrophilicity Index (eV) | Nucleophilicity Index (eV) |
|---|---|---|---|---|---|---|---|
| 8.0899 | 0.9163 | 4.5031 | −4.5031 | 7.1737 | 0.1394 | 1.4134 | 0.8681 |

c. Local Reactivity Descriptors for the Investigated Compound

The electrophilic $P_k^+$ and nucleophilic $P_k^-$ Parr functions, established by Domingo and his team, are useful local reactivity indicators that identify electrophilic and nucleophilic reactive sites in organic systems. These functions are based on the Atomic Spin Density (ASD) distribution in the radical anion and radical cation of neutral molecules. Interestingly, Domingo found that Parr functions provide more reliable reactivity outcomes compared to other commonly used local indicators like the Parr–Yang Fukui functions and the Yang–Mortier condensed Fukui functions.

In the case of the studied compound, the Parr functions were calculated, taking into account Atomic Spin Densities from the Mulliken population analysis, and the results are presented in Table 6 and illustrated in Figure 13. Upon analyzing the data, it was found that the maximum values of the electrophilic $P_k^+$ function are located over atoms C6, C2, C10, C12, C13, C14, C16, N62, N63, and S69. This suggests that these atoms are the most likely sites for nucleophilic attack, acting as electrophilic species in a chemical reaction involving this compound. On the other hand, the maximum values of the nucleophilic $P_k^-$ function are located over atoms C49, C52, C56, N65, and C59. This indicates that these atoms are more prone to electrophilic attacks, acting as nucleophilic species in chemical reactions.

3.4.2. Aromaticity and π–π Stacking Ability of the Tested Compound

The aromaticity and π–π stacking ability of the tested compound, a 2(5H)-furanone derivative, were quantitatively assessed using different electronic- and geometric-based aromaticity indices. The molecular structure was divided into various segments labeled as rings A, B, C, D, and E.

The Harmonic Oscillator Model of Aromaticity (HOMA) was used to assess the aromaticity of the compound. The HOMA indices for the benzene rings (A and B) were found to be 0.9891 and 0.989, respectively, indicating a strong aromatic character. Conversely, the HOMA value for the 2(5H)-furanone ring (D and E) was negative (−1.71464), suggesting an antiaromatic character [12].

The Bird Aromaticity Index, another measure of aromaticity, revealed that the benzene rings (A and B) exhibited a higher degree of aromaticity (96.93 and 96.16, respectively) compared to the 1,2,4-triazole ring (C) and the 2(5H)-furanone ring (D), which had Bird indices of 72.65 and 21.11, respectively (Figure 14 and Table 7).

The Shannon aromaticity index and the curvature of electron density, both measures of electron delocalization, support these findings. Rings A and B had lower Shannon indices (less than 0.003) and a more negative curvature of electron density, confirming their aromatic character. In contrast, rings C and D exhibited higher Shannon indices (0.004 and 0.006, respectively) and a less negative curvature of electron density, suggesting a lesser degree of aromaticity [47].

**Table 6.** Calculated local electrophilic $P_k^+$ and nucleophilic $P_k^-$ Parr functions, based on Mulliken Atomic Spin Densities for the examined compounds.

| Atom | $P_k^+$ | $P_k^-$ | Atom | $P_k^+$ | $P_k^-$ | Atom | $P_k^+$ | $P_k^-$ |
|------|---------|---------|------|---------|---------|------|---------|---------|
| Br1 | 0.9805 | 0.0061 | H24 | 0 | −0.0006 | H47 | 0 | 0 |
| C2 | 2.348 | 0.1731 | C25 | −0.0001 | 0.0012 | H48 | 0 | 0 |
| H3 | −0.1033 | −0.009 | H26 | 0 | 0.0005 | C49 | −0.0868 | 8.5184 |
| C4 | −1.4518 | −0.0231 | C27 | 0.0002 | 0.0029 | H50 | 0.0016 | −0.462 |
| H5 | 0.0451 | 0.0006 | H28 | 0 | −0.0038 | C51 | 0.0177 | 0.2508 |
| C6 | 5.3215 | 0.0529 | H29 | 0.0002 | 0.0002 | C52 | −0.0172 | 5.2379 |
| H7 | −0.2216 | −0.0013 | C30 | 0.0061 | 0.0036 | H53 | −0.0019 | −0.2616 |
| C8 | −1.7653 | 0.004 | H31 | −0.0003 | −0.0002 | C54 | 0.0085 | −2.859 |
| H9 | 0.0542 | −0.0005 | H32 | −0.0001 | −0.0003 | H55 | −0.0003 | 0.1008 |
| C10 | 2.6975 | 0.0162 | C33 | 0.0034 | −0.0016 | C56 | −0.0138 | 7.1603 |
| H11 | −0.1199 | −0.0014 | H34 | 0.0001 | −0.013 | C57 | 0.0085 | −2.1977 |
| C12 | 2.2075 | −0.0201 | C35 | 0.0377 | 0.0804 | H58 | −0.0003 | 0.0697 |
| C13 | 3.0258 | 0.3281 | H36 | −0.023 | 0.0701 | C59 | −0.0138 | 3.7211 |
| C14 | 3.9115 | 0.4641 | C37 | 0.0005 | −0.0017 | H60 | 0.0006 | −0.188 |
| C15 | −0.7243 | −0.0669 | H38 | 0.0002 | 0.0001 | Cl61 | −0.0013 | 0.105 |
| C16 | 2.0171 | 0.186 | H39 | −0.0017 | 0.0001 | N62 | 3.8215 | 0.1136 |
| C17 | −0.2322 | 0.0113 | H40 | 0.0003 | 0.0028 | N63 | 2.4328 | −0.0438 |
| C18 | 0.0083 | 0.0086 | C41 | 0.001 | 0.028 | N64 | −1.1932 | −0.3109 |
| H19 | −0.0139 | −0.0019 | H42 | 0.002 | 0.0127 | N65 | 0.0516 | 6.9202 |
| C20 | −0.0087 | −0.0052 | H43 | 0 | 0.0318 | O66 | 0.3651 | 0.0385 |
| H21 | 0.0005 | −0.0002 | H44 | 0.0002 | −0.0015 | O67 | 0.0522 | 0.0092 |
| C22 | 0.002 | −0.0011 | C45 | 0 | 0.0003 | O68 | 0.0249 | 0.0089 |
| H23 | −0.0001 | −0.0026 | H46 | 0 | 0.0001 | S69 | 3.7494 | −0.0499 |

Among the analyzed rings based on the LOLIPOP index (the Localized Orbital Locator Integrated Pi Over Plane) (Table 7), which indicates π-stacking ability, with lower values suggesting stronger π-depletion and enhanced values π-stacking propensity, Ring D exhibits the most significant π-stacking potential with a value of 0.085. This is followed by Ring C (0.159), indicating a moderate π-stacking tendency. Ring A, with a LOLIPOP value of 0.780, demonstrates a diminished π-stacking ability relative to Rings D and C, while Ring B, possessing the highest value of 2.019, exhibits the least propensity for π-stacking among the studied rings [12,47].

In the context of the aromatic compound under study, π depletion refers to a decrease in the delocalization of π electrons across the aromatic ring system. In contrast, π delocalization is a phenomenon where π electrons are spread over more than two atoms in a molecule, contributing to the stability and the aromatic nature of the compound. When there is higher π-depletion (or lower π-delocalization), it means that the π electrons are not as evenly spread out over the entire aromatic system. Instead, they are more localized to specific areas of the molecule. This scenario often occurs in compounds where the electron distribution is influenced by the presence of substituents or the geometry of the molecule, among other factors. A higher π-depletion can enhance the ability of the aromatic ring to engage in π-stacking interactions; π-stacking refers to attractive, noncovalent interactions between aromatic rings, where the electron-rich π-system of one molecule interacts with the π-system of another molecule. When the π electrons are more localized (higher

π-depletion), there may be regions of the molecule with higher electron density that can more effectively engage in these interactions [12].

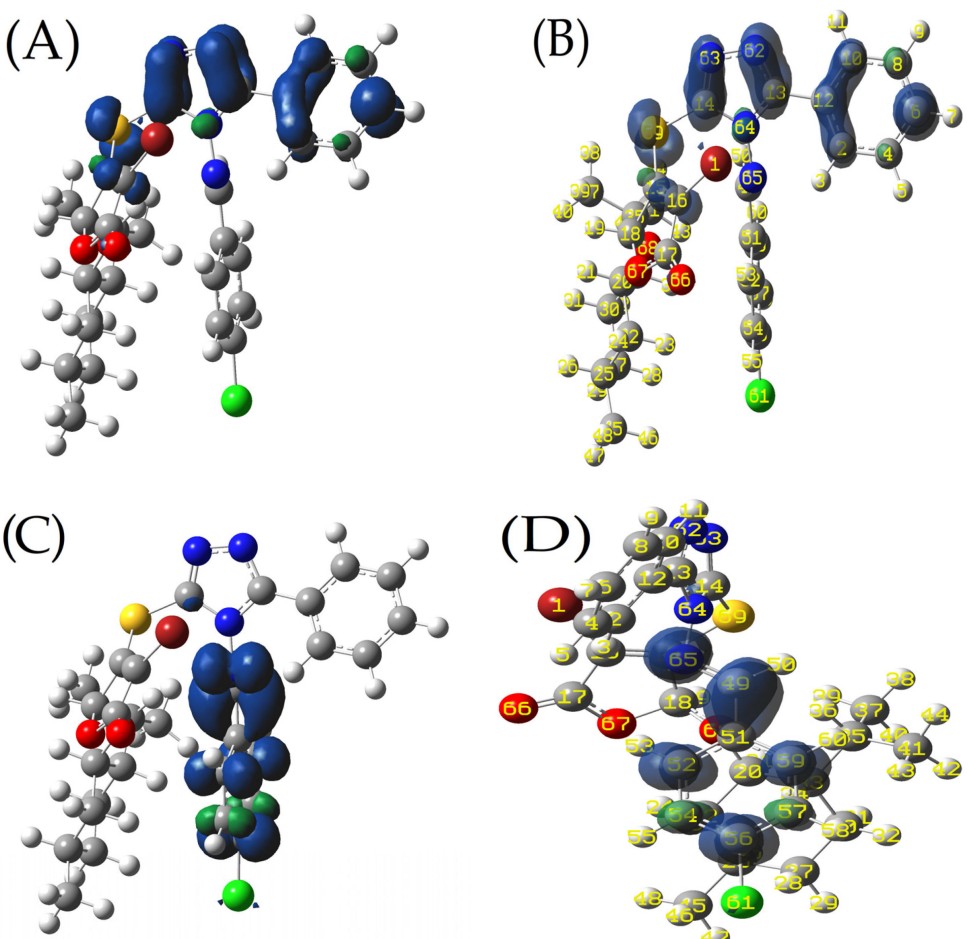

**Figure 13.** Visual representations of Mulliken Atomic Spin Densities for radical cations (**A**,**B**) and radical anions (**C**,**D**), accompanied by nucleophilic and electrophilic Parr functions for the investigated compound. Blue and green colors indicate positive and negative spin densities, respectively.

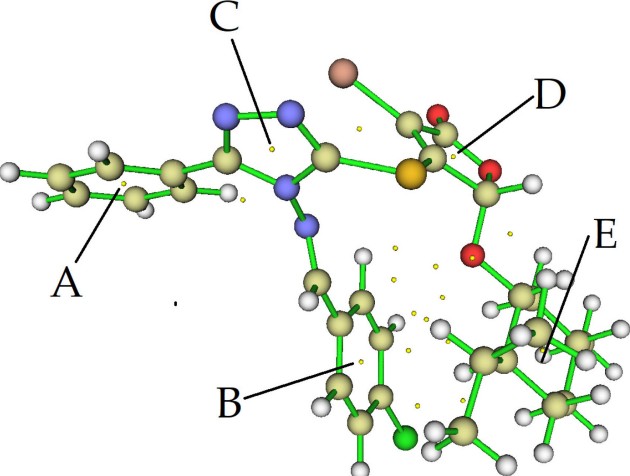

**Figure 14.** Depictions of molecular structures with ring and pseudoring labels, generated via MULTI-WFN. The labels correspond to different ring types within the structures: (A) represents a phenyl ring, (B) denotes a triazole ring, (C) corresponds to a chlorophenyl ring, (D) signifies an (S)-2-bromo-4-hydroxy-3-mercaptocyclopent-2-en-1-one ring, and (E) represents a cyclo-hexane ring.

**Table 7.** Calculated electronic and geometric aromaticity indices along with LOLIPOP measures for the analyzed compounds.

| Ring | Shannon Aromaticity | Curvature of Electron Density | HOMA | Bird Aromaticity Index | LOLIPOP Index |
|---|---|---|---|---|---|
| A | ≈0 | 0.003 | 0.989 | 96.93 | 0.780 |
| B | ≈0 | 0.008 | 0.989 | 96.16 | 2.019 |
| C | 0.004 | 0.004 | 0.866 | 72.65 | 0.159 |
| D | 0.006 | 0.005 | −1.715 | 21.11 | 0.085 |
| E | ≈0 | −0.014 | −4.285 | 97.46 | 0.000 |

In the examined compound, the five-membered ring (Ring D) appears to exhibit a pronounced propensity for π–π stacking interactions, as suggested by its low LOLIPOP index value of 0.084706 (Table 7) and the localization of its π electrons (double bond). This could imply a certain level of π depletion in this ring, thereby fostering stronger π–π stacking interactions [12,47].

The provided data of Ring E detail various indices related to cyclohexane's aromaticity and electronic characteristics. While cyclohexane is a non-aromatic six-membered aliphatic hydrocarbon, some of the values, notably the HOMA and LOLIPOP index, are consistent with its non-aromatic nature. However, the curvature of electron density and the Bird Aromaticity Index results seem anomalous, suggesting strong aromatic character, which contradicts the known properties of cyclohexane. However, the curvature of electron density and the Bird Aromaticity Index for Ring E deviate from expectations. It is well established that a negative curvature of electron density, especially a significant one, often indicates a strong aromatic character. Similarly, a high Bird Aromaticity Index is suggestive of aromaticity. The observed values for these indices, in the case of cyclohexane, are indeed anomalous.

Several potential factors might contribute to these unexpected results:

Molecular geometry: although cyclohexane is non-planar, slight deviations or perturbations in the molecule's geometry during the calculations might influence the derived electronic characteristics, leading to misleading aromaticity indices.

Inherent limitations: every index, regardless of its robustness, has inherent limitations. They might not always accurately reflect the true nature of a molecule, especially in edge cases or systems that deviate from standard aromatic compounds.

*3.5. The Molecular Electrostatic Potential (MEP)*

The Molecular Electrostatic Potential (MEP) of the compound (*E*)-3-bromo-4-((4-((1-(4-chlorophenyl)ethylidene)amino)-5-phenyl-4H-1,2,4-triazol-3-yl)thio)-5-((2-isopropylcyclohexyl)oxy)furan-2(5H)-one ($C_{29}H_{30}BrClN_4O_3S$) provides a critical understanding of its electrostatic properties and potential sites of chemical reactivity.

The MEP map is color-coded to visually depict the charge distribution across the molecule's surface (Figure 15A,B). The range of this color scale extends from −163.67 KJ/mol to +111.16 KJ/mol, with the most intense red representing the lowest negative MEP value and the deepest blue corresponding to the highest positive MEP value [29,48,49]. The blue regions on the MEP map, denoting positive MEP values, signify the electrophilic sites of the molecule. These sites are deficient in electrons and thus have a propensity to attract nucleophiles, which are electron-rich entities [48]. Notably, these electrophilic regions are primarily located around the hydrogen atoms bonded to the chloro-phenyl group, suggesting that these hydrogen atoms may be susceptible to nucleophilic attack [50]. In contrast, the red regions on the MEP map, associated with negative MEP values, indicate the molecule's nucleophilic sites. These sites have an abundance of electrons and are thus prone to attracting electrophiles [51]. The oxygen atoms of the carbonyl group and the nitrogen atoms of the triazol group are enveloped in these nucleophilic regions, implying

that these atoms could potentially be the sites of electrophilic attack [52]. The green regions on the MEP map, where the electrostatic potential is neutral, signify the areas of the molecule with a balanced charge distribution. These regions typically do not exhibit a significant excess or deficiency of electrons, indicating a stable state with less chemical reactivity [29,45,48,53–55].

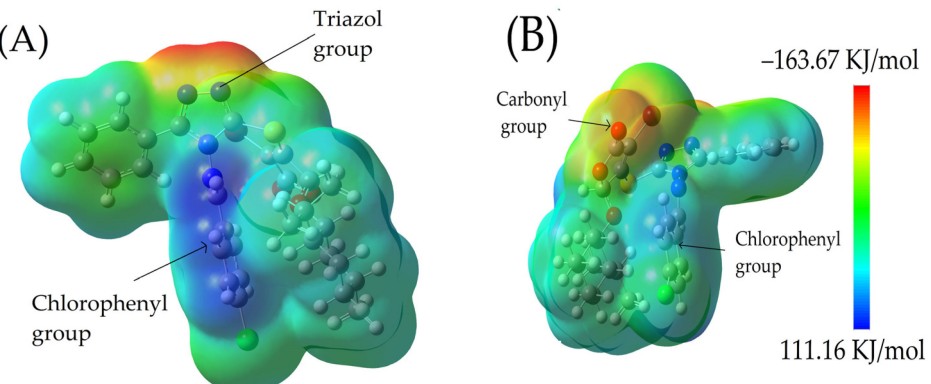

**Figure 15.** (**A**,**B**) Color-graded representation of the Molecular Electrostatic Potential map for the molecule, spanning from −163.67 KJ/mol to +111.16 KJ/mol.

## 4. Conclusions

The compound (*E*)-3-bromo-4-((4-((1-(4-chlorophenyl)ethylidene)amino)-5-phenyl-4H-1,2,4-triazol-3-yl)thio)-5-((2-isopropylcyclohexyl)oxy)furan-2(5H)-one unveils an intricate molecular architecture enriched by distinct functional groups, with the ethylidene bridge connecting the 4-chlorophenyl ring to the 1,2,4-triazole ring being a defining feature. Its crystalline parameters, characterized by the CCDC code 829447, encompass a monoclinic system within the *C*2 space group and a density of 1.405 g/cm$^3$.

When juxtaposing X-ray diffraction data with DFT evaluations, the high fidelity of the wB97XD functional emerges, marked by impressively minuscule MAEs for bond lengths and angles. The Hirshfeld surface analysis furnishes vital insights into the pivotal intermolecular interactions that dictate the crystal packing of this compound. The evident dominance of hydrogen bonding, exemplified by O1···H1 and C12···N1 interactions, underscores its importance within this molecular matrix. Notably, the absence of adjacent red and blue triangles in the Shape Index and the paucity of flat terrains in the Curvedness surface convincingly preclude the presence of π···π stacking interactions.

The analyses of d$_e$ and d$_i$ surfaces are instructive, depicting a judicious balance between internal and external molecular interactions, leading to a densely packed crystal configuration. The predominance of specific hydrogen bonds further corroborates the critical role of hydrogen bonding in the compound's structure.

The compound's aromatic attributes and π–π stacking capabilities were discerningly dissected. The Harmonic Oscillator Model of Aromaticity (HOMA) and Bird Aromaticity Index reinforce the aromatic nature of the benzene rings. In contrast, the 2(5H)-furanone ring exhibits an antiaromatic character. Rings A and B are indisputably aromatic, as affirmed by the Shannon aromaticity index and electron density curvature. The LOLIPOP index reveals the compelling π–π stacking potential of Ring D.

The Molecular Electrostatic Potential (MEP) offers a nuanced visualization of the molecule's electrostatic properties. Electrophilic regions, predominantly around the hydrogen atoms bonded to the chlorophenyl group, emerge as prime sites for nucleophilic interactions. Conversely, the oxygen atoms of the carbonyl group and the nitrogen atoms of the triazol group emerge as potential electrophilic interaction zones.

In essence, this research paints a comprehensive portrait of the compound's geometric, interactional, and reactivity nuances. This deep dive, harnessing both experimental and computational lenses, significantly augments our grasp of the compound's multifarious

molecular behavior and potential interactions, laying a robust groundwork for subsequent explorations or applications in diverse scientific arenas.

**Author Contributions:** A.H.B. was instrumental in the conceptualization of the study, as well as the development of the study methodology and software. They also carried out the validation, formal analysis, and investigation processes. Data curation and the drafting of the original manuscript were tasks undertaken by A.H.B., H.M.A. and A.A.K. The manuscript was then reviewed and edited by M.W.A., H.M.A. and A.H.B. Financial support for the study was secured by M.W.A. and A.A.K. All authors have read and agreed to the published version of the manuscript.

**Funding:** The Researcher Supporting Project Number (RSPD2023R760) at King Saud University, Riyadh, Saudi Arabia, has generously provided financial backing for this research.

**Data Availability Statement:** All data pertinent to this study can be found within the content of the manuscript.

**Acknowledgments:** The authors would like to extend their heartfelt appreciation to the Researcher Supporting Project Number (RSPD2023R760) at King Saud University, Riyadh, Saudi Arabia. The financial support provided by this project for the realization of this research is deeply appreciated.

**Conflicts of Interest:** The authors of this study wish to affirm that they do not hold any personal or financial relationships that could be misconstrued as a conflict of interest. The insights and viewpoints conveyed in this publication are the independent thoughts of the authors and should not be interpreted as mirroring the official policy or stance of the Department of Health and Human Services or any other governmental body. Additionally, any reference to commercial goods, institutions, or trade names within this publication is intended solely for informational relevance and does not imply any form of endorsement by the U.S. Government.

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
