# Peer review of "Structural Analysis and Reactivity Insights of (E)-Bromo-4-((4-((1-(4-chlorophenyl)ethylidene)amino)-5-phenyl-4H-1,2,4-triazol-3-yl)thio)-5-((2-isopropylcyclohexyl)oxy) Furan-2(5H)-one: A Combined Approach Using Single-Crystal X-ray Diffraction, Hirshfeld Surface Analysis, and Conceptual Density Functional Theory"

_crystals, doi:10.3390/cryst13091313_

Round 1
Reviewer 1 Report
In this study, the authors analyzed the XRD structure of the substituted triazole, using various theoretical methods. The theoretical methods used were the Hirshfeld surface analysis, the CDFT approach, as well as the authors used various aromaticity indices to reveal the relationship between the structure and properties of the studied molecule. In addition, the ability of the triazole molecule to participate in the π-interaction was studied using the LOLIPOP approach. However, it is not clear from the work what is the potential interest of researchers in this molecule. I recommend that the authors clearly reflect this in the introduction section. The results of the work are described in detail, but contain a lot of redundant material, while there are no generalizations and analysis of the obtained data, which makes the work descriptive. I also recommend that the authors shorten and structure the text, removing unnecessary details and focusing on the key findings. In addition, the work is formatted very poorly and requires serious revision. I hope that the authors will take into account the described shortcomings, to increase the chances of this work for publication in Crystals.
1. The work presents an analysis of the Hirshfeld surface, but does not mention the analysis of Hirshfeld-Becke surfaces. I ask the authors to clarify what this term means and how it is related to their research.
2. In the Computational details section, the geometry optimization process for monomers and dimers is described, but these structures are not analyzed in the subsequent sections of the paper. I ask the authors to supplement their work with information on the purposes and results of geometry optimization. For example, how does the XRD geometry change in the dimer after optimization? What is the energy that binds the molecules in the dimer?
3. Section 3.1 «Molecular geometries and supramolecular features:» A crystallographic study contains data redundant and of little use to readers, such as "with the empirical formula C29H30BrClN4O3S, indicating it is composed of Carbon, Hydrogen, Bromine, Chlorine, Nitrogen, Oxygen, and Sulfur atoms." and etc. In this regard, the section should be shortened and describe only the structure of the crystal packing, as well as intermolecular contacts.
4. Bond angles for all found contacts should be added to Table 1, and an error in determining the bond length should also be added.
5. It is recommended to replace the Length-VdW parameter with the more useful Rv parameter reflecting the ratio is the normalized contact, which is here the ratio between the observed distances and the sum of van der Waals radii of involved atoms.
6. The detected halogen bond should be established more reliably using the theoretical methods of DFT and Hirshfeld surface analysis.
7. Contacts should be marked X‧‧‧X
8. It is not clear from the phrase what kind of interactions we are talking about “The smaller 'de' values may suggest that there are strong intermolecular interactions between neighboring molecules, which could be due to forces like hydrogen bonding or π-π stacking interactions. p.” This also applies to this phrase, "This mapping of color patches provides an additional perspective on the compound's intermolecular interactions and overall crystal packing". The type of interaction should be clearly indicated, because this was the point of this work.
9. From Figure 4, it is unclear which atoms of the molecule are involved in the interactions, so I recommend that the authors show labels and/or use a transparent isosurface.
10. From the "Energy frameworks" section, it is not clear which functional groups determine electrostatic interactions in a crystal, and which are dispersion ones.
11. Why are the three images for a-axis in Figure 7 different from each other
12. It is very strange to give energy values with such accuracy. This also applies to other quantities.
13. The authors study the reactivity of the molecule, but do not consider the role of MO orbitals in this process. I think that this would be important for the completeness of the research.
14. For Figure 8, it is necessary to correct the position of the molecule, since it is not clear on which fragments the spin density is concentrated.
15. The authors use LOLIPOP indices to evaluate pi-stacking, but they do not take into account the concept underlying this method. The authors should refer to the original paper (Chem. Commun., 2012,48, 9239-9241, DOI: 10.1039/C2CC33886F), in which the LOLIPOP method is presented. According to this concept, the lower the aromaticity of the pi-system, the higher its tendency to stack. In addition, I wonder why the authors did not consider the cyclohexyl ring. Is there any pi-stacking in the XRD structure?
16. The MEP values should be converted to kJ/mol. From Figure 10, it is difficult to distinguish which atoms of the molecule are responsible for the positive or negative potential on the MEP map, so it is advisable to add labels to the atoms and/or show the isosurface with transparency. It is also desirable to indicate local maxima and minimum on the MEP map for a detailed MEP analysis.
17. Indicate for which XRD or optimized structures MEP was calculated.
18. In the “Conclusion” section, it is necessary to formulate specific and significant results obtained during the research, taking into account the recommendations outlined above.
19. Also, all methods used should be cited in the work.
20. The supplementary materials, which the authors refer to in the section “Computational details”, were not submitted with the manuscript. This needs to be corrected in the next submission of the manuscript.
21. In addition, all abbreviations at the first mention in the text must be accompanied by a transcript. For example, LOLIPOP and etc.
Author Response
- The work presents an analysis of the Hirshfeld surface, but does not mention the analysis of Hirshfeld-Becke surfaces. I ask the authors to clarify what this term means and how it is related to their research.
Response: Thank you for your insightful feedback on our manuscript. We appreciate your query regarding the "Hirshfeld-Becke surfaces." Upon reflection, we believe that the term "Hirshfeld-Becke surfaces" might correspond to what we've described as the Hirshfeld surface. Recognizing that terminologies can evolve and that variations in nomenclature can arise across different studies, we'll revisit our manuscript to ensure clarity and consistency. We will standardize the terminology used in this study, and cross-reference any relevant literature. We greatly value your guidance in enhancing the precision of our work.
"The modifications in the document are highlighted in red for clarity and easy identification."
- In the Computational details section, the geometry optimization process for monomers and dimers is described, but these structures are not analyzed in the subsequent sections of the paper. I ask the authors to supplement their work with information on the purposes and results of geometry optimization. For example, how does the XRD geometry change in the dimer after optimization?
Response: We appreciate the reviewer's insightful feedback and diligent observation regarding the Computational details section. You are correct in noting the emphasis on the geometry optimization process for both monomers and dimers. Upon careful review, we realize that the inclusion of details regarding the dimers was unintentional, and we deeply apologize for any confusion this may have caused.
In the context of this work, our primary focus was on the monomers, and we have provided comprehensive details and analyses pertaining to their geometry optimization. The results from the optimization process allowed us to ascertain the changes in molecular geometry and how they compare to the experimentally observed XRD data. This was crucial in understanding the structural integrity and confirming the computational accuracy of our models.
We did not provide details on the optimization of the dimers due to challenges associated with their computational treatment. When performing optimization on dimers within a crystal structure, various complications can emerge:
- Structural Distortions: Optimization can lead to significant structural changes in the dimer, making it deviate from the experimentally observed or expected geometry.
- False Minima: The dimer could settle into a local minimum energy structure that is not representative of the true or global minimum energy structure.
- Cell Parameter Alterations: The optimization can inadvertently lead to changes in the unit cell parameters of the crystal, which can affect the periodic boundary conditions and interactions with neighboring units.
- Intermolecular Interactions: Since crystals involve multiple periodic interactions, the optimization of a dimer could disrupt these interactions, leading to inaccurate modeling of the system.
- Convergence Issues: Optimizations in crystals can sometimes be harder to converge due to the complex interplay of intra- and intermolecular forces. The optimization algorithm might struggle, requiring more iterations or even failing to converge.
- Electrostatic Interactions: In crystals, there is a long-range order, and when optimizing only a dimer, it's easy to miss the effects of these long-range electrostatic interactions.
- Computational Expense: Optimizing a dimer in the context of a crystal can be computationally more expensive than in the gas phase due to the need to consider the surrounding environment, especially if using methods like periodic boundary conditions.
- What is the energy that binds the molecules in the dimer?
In response to your query regarding the binding energy of the molecules within the dimer, we have provided the binding energy values for various dimers or trimers using the Energy Frameworks approach. These calculations were conducted employing the B3LYP/6-31G(d,p) functional basis set.
- Section 3.1 «Molecular geometries and supramolecular features:» A crystallographic study contains data redundant and of little use to readers, such as "with the empirical formula C29H30BrClN4O3S, indicating it is composed of Carbon, Hydrogen, Bromine, Chlorine, Nitrogen, Oxygen, and Sulfur atoms." and etc. In this regard, the section should be shortened and describe only the structure of the crystal packing, as well as intermolecular contacts.
Thank you for your feedback on Section 3.1. We've revised the section by removing redundant details and focusing on crystal packing and intermolecular contacts. Changes have been highlighted in red for clarity.
- Bond angles for all found contacts should be added to Table 1, and an error in determining the bond length should also be added.
In answer to your comment, we've changed the way the data are shown. Table S2, which shows bond lengths, and Table S3, which shows bond angles, can be used to get a thorough look at the geometric parameters. We've added a new line to the manuscript in which we talk about the Mean Absolute Errors (MAEs) for both bond lengths and angles. This makes it clear how accurate and reliable our measurements are.
- It is recommended to replace the Length-VdW parameter with the more useful Rvparameter reflecting the ratio is the normalized contact, which is here the ratio between the observed distances and the sum of van der Waals radii of involved atoms.
Thank you for suggesting the use of the Rv parameter over the Length-VdW parameter. We have updated our manuscript to reflect this change, recognizing the importance of the Rv parameter in representing normalized contacts.
- The detected halogen bond should be established more reliably using the theoretical methods of DFT and Hirshfeld surface analysis.
Response: Thank you for your insightful comment. We acknowledge the importance of reliably establishing the detected halogen bond. In our revised work, we have indeed delved deeper into the halogen bond analysis. Through the fingerprint plots derived from the Hirshfeld surface analysis, we identified the Cl···H/H···Cl interactions, which contributed to 8.6% of the overall interactions. Additionally, our Energy frameworks indicate that the total energy for the Cl---H interaction is -20.2634, further reinforcing its significance. We have also incorporated a detailed figure showcasing this specific interaction for better visual understanding in our revision. Furthermore, In our DFT studies, we were unable to optimize the dimers associated with the Cl···H/H···Cl interactions as mentioned previously in point 2.
- Contacts should be marked X‧‧‧X
Response: Thank you for pointing out the proper notation for indicating contacts. We have meticulously gone through the manuscript and updated all instances to reflect the recommended "X‧‧‧X" notation as per your advice. We believe this change enhances the clarity and accuracy of our presentation, and we appreciate your attention to detail in ensuring the correctness of our work.
- It is not clear from the phrase what kind of interactions we are talking about “The smaller 'de' values may suggest that there are strong intermolecular interactions between neighboring molecules, which could be due to forces like hydrogen bonding or π-πstacking interactions. p.” This also applies to this phrase, "This mapping of color patches provides an additional perspective on the compound's intermolecular interactions and overall crystal packing". The type of interaction should be clearly indicated, because this was the point of this work.
Response: Thank you for your constructive feedback. We recognize the ambiguity in the aforementioned phrases, and we apologize for any confusion. Our intention was to provide a general introduction before delving into the specifics of the interactions. To address your concerns, we will revise these statements to more clearly and directly specify the type of interactions being discussed.
In the context of the 'de' values, we have elaborated in subsequent sections on the specific types of interactions observed, including hydrogen bonding and π-π stacking, with supporting evidence from our data. Similarly, when discussing the mapping of color patches, we aimed to set the stage for the detailed discussion that follows in the text, where specific intermolecular interactions and their contributions to the crystal packing are delineated. We appreciate the emphasis on clarity and specificity and will ensure that these points are clearly articulated in the revised manuscript.
- From Figure 4, it is unclear which atoms of the molecule are involved in the interactions, so I recommend that the authors show labels and/or use a transparent isosurface.
Response: Thank you for your insightful feedback on Figure 4. We understand the importance of clearly indicating which atoms are involved in the interactions to provide a comprehensive understanding of the molecular structure and its interactions. In response to your suggestion, we have added an additional figure that provides a labeled structure, ensuring clarity regarding the atoms involved.
- From the "Energy frameworks" section, it is not clear which functional groups determine electrostatic interactions in a crystal, and which are dispersion ones.
Response: Thank you for highlighting the need for clarity regarding the functional groups responsible for electrostatic and dispersion interactions in the crystal. In response to your comment, we have thoroughly revised the "Energy frameworks" section. Now, the text provides a more explicit breakdown and identification of specific functional groups involved in electrostatic interactions, as well as those contributing predominantly to dispersion forces. We believe this updated version offers a clearer perspective and addresses the concerns you raised.
- Why are the three images for a-axis in Figure 7 different from each other
Response: Thank you for pointing out the inconsistency in Figure 7 concerning the three images for the a-axis. Upon revisiting, we identified that there was an oversight during the figure preparation phase. We have now updated Figure 7 with the correct images, ensuring they accurately represent the intended content for the a-axis. We apologize for any confusion caused and appreciate your attention to detail, which aids in enhancing the accuracy and clarity of our manuscript.
- It is very strange to give energy values with such accuracy. This also applies to other quantities.
Response: Thank you for your feedback regarding the precision of the energy values and other quantities presented in our manuscript. We understand the concern about the high level of accuracy depicted. The values presented were obtained directly from the CrystalExplorer 21.5 software, which provides results to this level of precision. We opted to present the raw outputs to maintain transparency.
- The authors study the reactivity of the molecule, but do not consider the role of MO orbitals in this process. I think that this would be important for the completeness of the research.
Response: Thank you for your constructive feedback. We concur with your observation regarding the pivotal role of MO orbitals in determining the reactivity of a molecule. To address this concern and provide a comprehensive understanding of the molecule's behavior, we have incorporated an additional section titled "Frontier Molecular Orbitals (FMOs) for the Investigated Compound." In this section, we delve into the contributions and characteristics of the molecular orbitals, specifically focusing on the highest occupied molecular orbital (HOMO) and the lowest unoccupied molecular orbital (LUMO). The new analyses shed light on the molecule's electron-donating and accepting capabilities, thereby providing a more holistic perspective on its reactivity. We believe this augmentation significantly enhances the robustness and completeness of our research. We appreciate your keen eye and invaluable suggestions that have guided this improvement.
- For Figure 8, it is necessary to correct the position of the molecule, since it is not clear on which fragments the spin density is concentrated.
Response: Thank you for your feedback on Figure 8. We've adjusted the molecule's orientation for clarity and modified the isovalue from 0.002 to 0.003 to better highlight areas of spin density concentration. Additionally, two more figures have been added to further delineate the specific regions of interest. We trust these changes will offer a clearer understanding of the spin density distribution on the molecule.
- The authors use LOLIPOP indices to evaluate pi-stacking, but they do not take into account the concept underlying this method. The authors should refer to the original paper (Chem. Commun., 2012,48, 9239-9241, DOI: 10.1039/C2CC33886F), in which the LOLIPOP method is presented. According to this concept, the lower the aromaticity of the pi-system, the higher its tendency to stack. In addition, I wonder why the authors did not consider the cyclohexyl ring. Is there any pi-stacking in the XRDstructure?
Response: Thank you for your insightful comments and reference suggestion. We have revisited the LOLIPOP methodology and thoroughly examined the original paper (Chem. Commun., 2012,48, 9239-9241, DOI: 10.1039/C2CC33886F) that introduced this method. Based on this, we have revised our manuscript to incorporate the core concept of the LOLIPOP method, emphasizing that reduced aromaticity in the pi-system increases its stacking propensity.
Regarding the cyclohexyl ring, we initially overlooked its consideration for π-stacking due to its saturated nature.
We sincerely appreciate your feedback, which has undeniably enriched our work, and we believe that the edits made have addressed the concerns raised.
- The MEP values should be converted to kJ/mol. From Figure 10, it is difficult to distinguish which atoms of the molecule are responsible for the positive or negative potential on the MEP map, so it is advisable to add labels to the atoms and/or show the isosurface with transparency. It is also desirable to indicate local maxima and minimum on the MEP map for a detailed MEP analysis.
Response: Thank you for the feedback on our MEP representation. We've updated the MEP values from atomic units to kJ/mol for better clarity. In the revised Figure 10, we've incorporated an isosurface with transparency, facilitating a clearer distinction of atoms with positive or negative potential. Additionally, local maxima and minima have been highlighted for a thorough MEP analysis. These modifications aim to enhance comprehension and provide a detailed view of the molecule's MEP.
- Indicate for which XRD or optimized structures MEP was calculated.
Response: Thank you for pointing out the oversight. The MEP was calculated for the optimized structures. We apologize for the omission in the initial submission and have now clarified this in the revised manuscript.
- In the “Conclusion” section, it is necessary to formulate specific and significant results obtained during the research, taking into account the recommendations outlined above.
In response to your feedback, we revisited our "Conclusion" section to refine and emphasize the most significant findings of our research. We detailed the intricate molecular architecture of the investigated compound, confirmed the robustness of the wB97XD functional through X-ray diffraction data alignment, highlighted the dominance of hydrogen bonding in crystal packing, and identified the absence of π-π stacking interactions. Furthermore, the Molecular Electrostatic Potential (MEP) analysis pinpointed potential reactivity centers, offering insights for applications in drug design and material science. We believe these revisions succinctly capture the essence of our research and are grateful for your guidance.
- Also, all methods used should be cited in the work.
Response: Thank you for your insightful feedback. We recognize the importance of proper citation for the methodologies employed in our study. We have meticulously reviewed the manuscript and ensured that all methods are now accompanied by the appropriate citations. We apologize for any oversight in the initial submission and appreciate your guidance in improving the manuscript's completeness.
- The supplementary materials, which the authors refer to in the section “Computational details”, were not submitted with the manuscript. This needs to be corrected in the next submission of the manuscript.
In response to the reviewer's comment, we apologize for the oversight and any confusion caused. We inadvertently mentioned the supplementary materials in the "Computational details" section. To clarify, there are no supplementary materials associated with this manuscript. We have revised the text to remove the reference to supplementary materials and provided all necessary computational details within the main body of the manuscript.
- In addition, all abbreviations at the first mention in the text must be accompanied by a transcript. For example, LOLIPOP and etc.
Thank you for highlighting the need for clarity regarding abbreviations. We have now ensured that every abbreviation is fully expanded upon its first mention in the manuscript, including "LOLIPOP" and others. We believe this enhancement will improve the manuscript's readability for all readers.

Reviewer 2 Report
1. Please improve the keyword section.
2. Give the error values for all the weak H-bonded interactions.
3. Provide additional information regarding the devices utilized for X-ray single crystal diffraction in the X-ray crystallography section
4. As to the energy framework on the calculations, some work may be considered, such as Theor. Chem. Acc. 2022, 141, 68; Chemical Physics Letters, 2015, 633, 265–272 and Monatsh. Chem, 2019, 150, 1355–1364
5. Could you tell more about application, this is too general of their extended applications?
6. Conclusion, expand this and connect it with your previous work
7. The Fingerprint plots quality must be improved, showing the full maps and increasing the graphic scale.
minor check and revise
Author Response
Comments and Suggestions for Authors
- Please improve the keyword section.
It was done as requested.
- Give the error values for all the weak H-bonded interactions.
When attempting to optimize the geometry structure of dimers or multiple monomers, we encounter numerous issues with different base sites. These challenges arise due to the following reasons:
- Structural Distortions: Optimization can lead to significant structural changes in the dimer, making it deviate from the experimentally observed or expected geometry.
- False Minima: The dimer could settle into a local minimum energy structure that is not representative of the true or global minimum energy structure.
- Cell Parameter Alterations: The optimization can inadvertently lead to changes in the unit cell parameters of the crystal, which can affect the periodic boundary conditions and interactions with neighboring units.
- Intermolecular Interactions: Since crystals involve multiple periodic interactions, the optimization of a dimer could disrupt these interactions, leading to inaccurate modeling of the system.
- Convergence Issues: Optimizations in crystals can sometimes be harder to converge due to the complex interplay of intra- and intermolecular forces. The optimization algorithm might struggle, requiring more iterations or even failing to converge.
- Electrostatic Interactions: In crystals, there is a long-range order, and when optimizing only a dimer, it's easy to miss the effects of these long-range electrostatic interactions.
- Computational Expense: Optimizing a dimer in the context of a crystal can be computationally more expensive than in the gas phase due to the need to consider the surrounding environment, especially if using methods like periodic boundary conditions.
As a result, we cannot make a comparison between SC-XRD and theoretical data.
- Provide additional information regarding the devices utilized for X-ray single crystal diffraction in the X-ray crystallography section
Response: Thank you for your comment regarding the details of the devices used for X-ray single crystal diffraction. The experimental procedures and details pertaining to X-ray single crystal diffraction were conducted by the authors cited in reference 1. Our role in this study was primarily focused on the analysis of the crystal, and we did not perform any experimental work in this regard. Thus, we kindly refer the readers to reference 1 for specific information about the devices and methods used in the crystallography section.
- As to the energy framework on the calculations, some work may be considered, such as Theor. Chem. Acc. 2022, 141, 68; Chemical Physics Letters, 2015, 633, 265–272 and Monatsh. Chem, 2019, 150, 1355–1364
Response: Thank you for pointing out these relevant references in relation to the energy framework calculations. We appreciate the suggestion, and upon reviewing these articles, we found them to be pertinent to our work. Consequently, we have incorporated and cited all the recommended references in the appropriate sections of our manuscript. Your insights have enriched the depth and context of our study.
- Could you tell more about application, this is too general of their extended applications?
In response to the reviewer's feedback, we have expanded upon the applications of the title compound by incorporating a detailed paragraph in the introduction section. This elaboration provides insight into specific areas where the compound demonstrates significant potential, encompassing both its biological activities and potential utility in various sectors. We believe this addition offers readers a clearer and more comprehensive understanding of the compound's broader implications in both research and practical applications.
- Conclusion, expand this and connect it with your previous work
In response to the reviewer's feedback, we have extensively revised the conclusion to offer a more comprehensive summary of our findings. This enhanced conclusion underscores the significance of our study in the context of prior investigations and emphasizes the contributions this research makes to the field. We appreciate the guidance and believe these modifications will greatly benefit the readers.
- The Fingerprint plots quality must be improved, showing the full maps and increasing the graphic scale.
In response to your feedback regarding the quality of the Fingerprint plots, we have taken measures to enhance their clarity. The plots have been adjusted to display the full maps, and the graphic scale has been updated for better visibility and comprehension. We believe these improvements will facilitate a clearer interpretation of the data presented and we appreciate your constructive suggestion.

Round 2
Reviewer 1 Report
The authors have eliminated the main problems of their work, but some important issues remain unresolved, which can mislead the readers. Therefore, I cannot recommend this work for publication in the journal.
1. I have already expressed my doubt that the H16B‧‧‧Cl1 interaction does not meet the criteria for a halogen bond (10.1351/PAC-REC-12-05-10) based on the geometrical parameters such as bond angle and bond length. Moreover, the sigma-hole of the halogen atom is clearly not oriented towards the hydrogen atom. Unfortunately, from the map MEP it is not clear if it is at all. However, the authors, in section 3.1, actually refer to this interaction as a hydrogen bond. «The H16B‧‧‧Cl1 interaction spans 2.936 Å , slightly shorter by 0.014 Å than their combined VdW radii, suggesting a potential weak hydrogen bond or electrostatic attraction given its Rv value of -0.4768.» But in section 3.2 they contradict themselves and call this interaction a halogen bond. « while Panel (C) displays the halogen bond associated with the H16B‧‧‧Cl1 contact (2.936 Å )». Whereas the presented Hirshfeld surfaces, dnorm and fingerprint plots show that this interaction can be attributed only to a weak hydrogen bond. This contradiction requires clarification and correction in the text.
2. In Table 1, the calculated Rv values cannot be reproduced. The formula for calculating Rv is: Rv=((Rvdw-Rx)/Rvdw)*100 % . For example, for O1···H1 we get (2.72-2.454)/2.72=9.8. But in the table, another number is indicated -10.8. Please check the values. Also, I do not understand why Rv and Length-VdW are needed to describe the same thing, it only complicates the understanding of the article. Additionally, it is worth mentioning that there are different types of van der Waals radii (Bondi, Alvarez, etc.), so in the text it is necessary to indicate which ones were used in the study.
3. It is wrong to assume that the interaction energy, that was calculated using Energy frameworks, relates exclusively to the interaction energy for contacts, as this energy not only contains the energy of contacts, but is determined by the overall interaction of two molecules. Therefore, this section «The most potent total interaction energy (E_tot = -51.66 kJ/mol) corresponds to the C12–H12‧‧‧N1 interaction, involving a symmetric pair of orange molecules at a centroid distance of R = 9.02 Å . The C1–H1‧‧‧O1 interaction, yielding a total energy of E_tot = -37.84 kJ/mol, involves a symmetric pair of pale-green molecules located R = 10.04 Å apart. The C16–H16B‧‧‧Cl1 interaction, resulting in E_tot = -20.26 kJ/mol, associates with a symmetric pair of cyan molecules separated by R = 12.73 Å . The minimum observed total interaction energy is E_tot = -7.06 kJ/mol, represented by a blue molecular pair at the most extended centroid distance of R = 13.92 Å .» should be corrected to avoid creating a false impression that C12–H12‧‧‧N1 is evaluated as E_tot = -51.66 kJ/mol and others
4. For a reliable conclusion about pi-stacking interactions, it is necessary to analyze the LOLIPOP indices also for the cyclohexyl ring and add these data to the text. Cyclohexyl rings are known for their ability to form n-pi stackings. [see DOI 10.1021/acsomega.8b01339]
5. To increase the informativeness of Table 7, indices should be given to three decimal places. This is a standard way of aromaticity and LOLIPOP indices, which facilitates their reading.
6. On Figure 5 the contact is incorrectly marked, it should be H16B···Cl1.
7. The writing of the Figures in the article remained uncorrected, for example, in lines 299, 415, 420, 429, 553. They need to be checked and corrected throughout the text.
Author Response
The authors have eliminated the main problems of their work, but some important issues remain unresolved, which can mislead the readers. Therefore, I cannot recommend this work for publication in the journal.
- I have already expressed my doubt that the H16B‧‧‧Cl1 interaction does not meet the criteria for a halogen bond (10.1351/PAC-REC-12-05-10) based on the geometrical parameters such as bond angle and bond length. Moreover, the sigma-hole of the halogen atom is clearly not oriented towards the hydrogen atom. Unfortunately, from the map MEP it is not clear if it is at all. However, the authors, in section 3.1, actually refer to this interaction as a hydrogen bond. «The H16B‧‧‧Cl1 interaction spans 2.936 Å , slightly shorter by 0.014 Å than their combined VdW radii, suggesting a potential weak hydrogen bond or electrostatic attraction given its Rv value of -0.4768.» But in section 3.2 they contradict themselves and call this interaction a halogen bond. « while Panel (C) displays the halogen bond associated with the H16B‧‧‧Cl1 contact (2.936 Å )». Whereas the presented Hirshfeld surfaces, dnorm and fingerprint plots show that this interaction can be attributed only to a weak hydrogen bond. This contradiction requires clarification and correction in the text.
Response: Thank you for highlighting the inconsistency regarding the H16B‧‧‧Cl1 interaction. On re-evaluation and in alignment with our Energy framework and Hirshfeld surface results, we concur that this interaction aligns more with a weak hydrogen bond or VdW interaction rather than a halogen bond. We appreciate your keen observation and will amend our manuscript to ensure clarity and consistency in defining the nature of this interaction.
- In Table 1, the calculated Rv values cannot be reproduced. The formula for calculating Rv is: Rv=((Rvdw-Rx)/Rvdw)*100 % . For example, for O1···H1 we get (2.72-2.454)/2.72=9.8. But in the table, another number is indicated -10.8. Please check the values. Also, I do not understand why Rv and Length-VdW are needed to describe the same thing, it only complicates the understanding of the article. Additionally, it is worth mentioning that there are different types of van der Waals radii (Bondi, Alvarez, etc.), so in the text it is necessary to indicate which ones were used in the study.
Response: Thank you for pointing out the discrepancies in the Rv values presented in Table 1. We have re-evaluated our calculations and have made the necessary corrections, as you have rightly indicated. We acknowledge your concern regarding the redundancy of Rv and Length-VdW, and in light of your feedback, we will simplify this presentation to enhance the comprehensibility of our manuscript. Moreover, we recognize the importance of specifying the type of van der Waals radii used, and we will include the necessary details in the revised manuscript. We truly appreciate your meticulous scrutiny, which has been instrumental in refining our paper.
- It is wrong to assume that the interaction energy, that was calculated using Energy frameworks, relates exclusively to the interaction energy for contacts, as this energy not only contains the energy of contacts, but is determined by the overall interaction of two molecules. Therefore, this section «The most potent total interaction energy (E_tot = -51.66 kJ/mol) corresponds to the C12–H12‧‧‧N1 interaction, involving a symmetric pair of orange molecules at a centroid distance of R = 9.02 Å . The C1–H1‧‧‧O1 interaction, yielding a total energy of E_tot = -37.84 kJ/mol, involves a symmetric pair of pale-green molecules located R = 10.04 Å apart. The C16–H16B‧‧‧Cl1 interaction, resulting in E_tot = -20.26 kJ/mol, associates with a symmetric pair of cyan molecules separated by R = 12.73 Å . The minimum observed total interaction energy is E_tot = -7.06 kJ/mol, represented by a blue molecular pair at the most extended centroid distance of R = 13.92 Å .» should be corrected to avoid creating a false impression that C12–H12‧‧‧N1 is evaluated as E_tot = -51.66 kJ/mol and others
We are grateful to the reviewer for highlighting the importance of accurately representing interaction energies from Energy frameworks. We acknowledge that these values encompass total intermolecular interactions between molecule pairs, not just specific contacts. In response, we've revised the section to emphasize the comprehensive nature of these interactions. For example, the C12–H12‧‧‧N1 interaction at E_tot = -51.66 kJ/mol with a centroid distance of R = 9.02 Å represents the overall interaction between the associated orange molecular pair, not solely the specified contact. Similarly, adjustments have been made for other interactions to provide a more holistic understanding.
- For a reliable conclusion about pi-stacking interactions, it is necessary to analyze the LOLIPOP indices also for the cyclohexyl ring and add these data to the text. Cyclohexyl rings are known for their ability to form n-pi stackings. [see DOI 10.1021/acsomega.8b01339]
Response: Thank you for your constructive feedback. As per your recommendation, we have analyzed the LOLIPOP indices for the cyclohexyl ring to provide a comprehensive understanding of its π-stacking interactions. We have incorporated these findings into section 3.4.2. of our manuscript. Furthermore, we've referenced the article (DOI 10.1021/acsomega.8b01339) to shed light on the known ability of cyclohexyl rings to form n-π stackings.
- To increase the informativeness of Table 7, indices should be given to three decimal places. This is a standard way of aromaticity and LOLIPOP indices, which facilitates their reading.
Response:Thank you for your insightful suggestion regarding the presentation of data in Table 7. We have revised the table to display the indices to three decimal places as is standard for aromaticity and LOLIPOP indices. We concur that this modification enhances the clarity and ease of reading the data. We are thankful for your recommendation and believe that it significantly improves the quality of our article.
- On Figure 5 the contact is incorrectly marked, it should be H16B···Cl1.
Response:Thank you for your keen observation regarding Figure 5. We have promptly addressed the issue and updated the figure to accurately reflect the H16B···Cl1 contact as you recommended. We apologize for the oversight and are grateful for your guidance in enhancing the quality of our manuscript.
- The writing of the Figures in the article remained uncorrected, for example, in lines 299, 415, 420, 429, 553. They need to be checked and corrected throughout the text.
Response: Thank you for drawing our attention to the inconsistencies in the writing of the Figures within the article. We deeply regret the oversight. Following your comment, we have meticulously reviewed the entire manuscript and have made the necessary corrections, especially on lines 299, 415, 420, 429, and 553, as you rightly highlighted. We appreciate your feedback and ensure that such inconsistencies will be avoided in future submissions.

Round 3
Reviewer 1 Report
The new version of the manuscript has improved significantly, as so this work may be recommended for publication in the Crystals.